# Associations of smoking and alcohol consumption with healthy ageing: a systematic review and meta-analysis of longitudinal studies

Christina Daskalopoulou,[1] Brendon Stubbs,[2,3] Carolina Kralj,[1] Artemis Koukounari,[4] Martin Prince,[1] A. Matthew Prina[2]

[1]Health Service and Population Research Department, Centre for Global Mental Health, Institute of Psychiatry, Psychology and Neuroscience, King's College London, London, UK
[2]Health Service and Population Research Department, Institute of Psychiatry, Psychology and Neuroscience, King's College London, London, UK
[3]Physiotherapy Department, South London and Maudsley NHS Foundation Trust, London, UK
[4]Department of Clinical Sciences, Liverpool School of Tropical Medicine, Liverpool, UK

**Correspondence to**
Mrs Christina Daskalopoulou; christina.daskalopoulou@kcl.ac.uk

## ABSTRACT

**Objectives** The number of older people is growing across the world; however, quantitative synthesis of studies examining the impact of lifestyle factors on the ageing process is rare. We conducted a systematic review and meta-analysis of longitudinal studies to synthesise the associations of smoking and alcohol consumption with healthy ageing (HA).

**Methods** Major electronic databases were searched from inception to March 2017 (prospectively registered systematic reviews registration number CRD42016038130). Studies were assessed for methodological quality. Random-effect meta-analysis was performed to calculate pooled ORs and 95% CI.

**Results** In total, we identified 28 studies (n=184 543); 27 studies reported results on smoking, 22 on alcohol consumption. 23 studies reported a significant positive association of never or former smoking with HA and 4 non-significant. 12 studies reported a significant positive association of alcohol consumption with HA, 9 no association and 1 negative. Meta-analysis revealed increased pooled OR of HA for never smokers compared with current smokers (2.36, 95% CI 2.03 to 2.75), never smokers compared with former smokers (1.32, 95% CI 1.23 to 1.41), former or never smokers compared with current smokers (1.72, 95% CI 1.20 to 2.47), never smokers compared with past or current smokers (1.29, 95% CI 1.16 to 1.43); drinkers compared with non-drinkers (1.28, 95% CI 1.08 to 1.52), light drinkers compared with non-drinkers (1.12, 95% CI 1.03 to 1.22), moderate drinkers compared with non-drinkers (1.35, 95% CI 0.93 to 1.97) and high drinkers compared with non-drinkers (1.25, 95% CI 1.09 to 1.44). There was considerable heterogeneity in the definition and measurement of HA and alcohol consumption.

**Conclusions** There is consistent evidence from longitudinal studies that smoking is negatively associated with HA. The associations of alcohol consumption with HA are equivocal. Future research should focus on the implementation of a single metric of HA, on the use of consistent drinking assessment among studies and on a full-range of confounding adjustment. Our research also highlighted the limited research on ageing in low-and-middle-income countries.

## Strengths and limitations of this study

► This is the first systematic review and meta-analysis trying to quantify the associations of smoking and alcohol consumption with healthy ageing; our findings were also adjusted for publication bias.

► Independent double screening, searching of previous systematic reviews in the field and the reference lists of the eligible studies as well as not excluding non-English studies guarantee that limited number of studies may not have been considered.

► Heterogeneity in the definition and measurement of healthy ageing as well as different confounding adjustment in the initial studies may have influenced our meta-analytical results.

► The number of studies included in the meta-analysis regarding the associations of alcohol consumption was small due to the great heterogeneity in its measurement.

► Our findings may not be generalisable due to the limited research in low-and-middle-income countries.

## INTRODUCTION

Smoking has emerged as the leading contributing factor of many respiratory diseases and one of the most important risk factors for cardiovascular diseases, cancers of several organs and many other pathological conditions.[1] In systematic reviews focusing on older people, smoking is associated with premature mortality[2] and an increased incidence of Alzheimer's disease.[3] Smoking is also linked with a considerable societal cost as smoking-related absenteeism and premature deaths create an additional burden to the healthcare system and to the economy in general.[4] The healthcare cost of tobacco-related diseases accounted for almost 6% of the global health expenditures in 2012.[5]

Tobacco use and harmful use of alcohol are recognised by the United Nations Political Declaration on Non-Communicable Diseases among the most common risk

factors.[6] Harmful use of alcohol results in approximately 3.3 million deaths annually (5.9% of all deaths) and alcohol-attributable deaths are the highest for middle-aged people.[7] Older people's sensitivity to the effects of alcohol increases whereas tolerance decreases,[8] meaning that even a low amount of alcohol can create problems in an older person. In addition, older people who drink harmfully often do not present to services due to stigma associated with harmful drinking, and when they do, their symptoms can be easily confused with physical and mental issues associated with ageing.[9] Nevertheless, studies showed that light to moderate alcohol consumption is associated with a decreased risk in dementia,[10] as well as with a reduced risk for cardiovascular disease mortality, incident coronary heart disease, incident stroke, stroke mortality and all-cause mortality.[11]

Drinking and smoking are correlated in the general population[12] with smokers being more likely to drink and vice versa.[13] People who are smokers and heavy drinkers exhibit the highest mortality rate as indicated in a study of middle-aged male participants[14]; another study also showed that men who smoked and drank >15 units per week of alcohol had almost three times higher all-cause mortality rate compared with non-smokers and non-drinkers.[15] As populations are ageing rapidly, with one in eight people currently aged ≥60 and with these estimates predicted to increase,[16] it is critical to understand the factors that will enable people to live longer and age in a healthy way.

To address the need for more accurate estimates of the relationship between smoking and healthy ageing and alcohol consumption and healthy ageing, we conducted a systematic review and meta-analysis of longitudinal studies which examined the associations of these factors with the ageing process.

## METHODS
### Search strategy and selection criteria
This systematic review has been written following the Preferred Reporting Items for a Systematic Review and Meta-Analysis (PRISMA) and the guidelines for Meta-Analyses Of Observational Studies in Epidemiology to ensure accuracy and comprehensiveness.[17 18] (online supplementary appendix 1). A review protocol was prospectively registered in the international database of prospectively registered systematic reviews under registration number CRD42016038130 (online supplementary appendix 2). As part of a larger body of work considering modifiable lifestyle factors and healthy ageing, we originally planned to carry out a review focusing on the following: physical activity, smoking and alcohol consumption. The current systematic review specifically focuses on the findings related to smoking, alcohol consumption and healthy ageing outcomes since a sufficient amount of literature was identified on this topic alone. Our findings regarding healthy ageing and physical activity have been presented elsewhere.[19]

Two authors (CD, CK) searched MEDLINE (PubMed/PubMed Central interface), EMBASE (OVID interface), PsycINFO (OVID interface) and CENTRAL from inception up to April 2016. Searching methodology included any related term or synonym to healthy ageing and text word related to physical activity, smoking and alcohol consumption (online supplementary appendix 3). Other relevant systematic reviews and reference lists of the eligible studies were also searched. Finally, a second search was performed on 15th March 2017 to include studies that were recently published.

Eligible studies had to meet the following criteria: (i) being published in an electronic journal article, (ii) constitute an original peer-reviewed longitudinal study and (iii) report any kind of longitudinal association between smoking and/or alcohol consumption and healthy ageing. Data on smoking and alcohol, through self-report and/or specific questionnaires, referred either to the current status and/or across the life span. The primary outcome of this review was health status measured by healthy ageing and any other term related to it (eg, successful ageing, active ageing, healthy survival). Studies whose primary goal was the examination of a different determinant but included the aforementioned factors as covariates were also included. Due to the heterogeneity of the healthy ageing definition, studies reporting the latter as multiple outcomes or based solely on self-report were excluded. Studies that included cohorts of individuals who were institutionalised or hospitalised and animal studies were also excluded. No language restriction was applied.

In order to store the studies that were retrieved by the electronic search, we used an EndNote (ENDNOTE X7, Thomson Reuters) library. After removal of duplicates, the two independent reviewers screened titles and abstracts of all potentially eligible papers. At the end of this procedure, any disagreement was solved by discussion. In case that an agreement could not be achieved, eligibility of the study was judged by discussion with a third senior researcher (AMP). In case that full text could not be retrieved, the corresponding author of the paper was contacted via email.

### Data extraction and quality assessment
CD and CK independently extracted data from each study and a random sample of them was cross-checked by AMP. Setting/country of the study, data collection period, follow-up year, sample size, population and baseline age information was recorded for each study. Definition and measurement of the healthy ageing outcome, of smoking and alcohol consumption, were also recorded, as well as the Odds Ratios (ORs) or any other related statistic and the 95% confidence intervals (CIs). Reported statistics were extracted for the least and most adjusted models reported in the study.

The methodological quality of the included studies was assessed by using the Quality in Prognosis Studies (QUIPS) tool. QUIPS evaluates six potential components

of bias: inclusion, attrition, prognostic factor measurement, confounders, outcome measurement, and analysis and reporting.[20] During the application of the QUIPS tool, smoking and/or alcohol consumption were considered as the only prognostic factors and all other variables, used as explanatory variables of the model, were considered as confounders. Since we included only longitudinal studies, the existence of attrition was expected. In the scenario that the attrition rate was high, authors' explanations were required to evaluate the study as bearing low risk of bias. Finally, the reliability of statistical models was evaluated according to the data presented; for instance, the reliability of studies that included results only for the statistically significant factors was judged with caution.

### Patient and public involvement

No patients were involved in this study. We used data from published papers only.

### Data analysis

We performed meta-analyses in order to produce a pooled effect size estimate for the relationship between smoking and healthy ageing and alcohol consumption and healthy ageing. Separate meta-analyses were performed for the following categories.

#### Smoking

(i) Past and never smokers compared with current smokers, (ii) former or never smokers compared with current smokers and (iii) never smokers compared with former or current smokers.

#### Alcohol

For studies that provided alcohol quantities, we converted grams to drinks by assuming that 12 g of alcohol is approximately one drink to have an alcohol consumption variable comparable among studies.[21] In studies that provided alcohol consumption in an interval, we took the middle of the interval as the most likely value. We created the following categories: light (<1 drink/day), moderate (1–2 drinks/day) and high (>2–4 drinks/day) consumption that were compared with never drinkers. Studies that did not report specific alcohol quantity but characterised the participants as drinkers or non-drinkers were examined separately.

We produced pooled effect estimates of the aforementioned categories for the most adjusted models. Due to the expected heterogeneity, a random-effects meta-analysis using the DerSimonian-Laird model was performed.[22] If a study reported different results for men and women, both results were included, except in cases where a result for the mixed population was also provided; the same rationale was applied to cases that reported statistics per different subgroups in the cohort. We also investigated via subgroup analyses the effect of baseline age (ie, studies with baseline mean age >65 years and studies with baseline mean age ≤65 years) and follow-up time (ie, studies with follow-up time >10 years and studies with follow-up time ≤10 years) to the pooled effect estimates. Heterogeneity was assessed with I[2] statistic for each analysis,[23] and publication bias[24] was assessed with Egger bias test.[25] Finally, a trim-and-fill adjusted analysis was conducted to adjust for potential publication bias.[26] Analyses were performed in STATA V.14 IC.

### Results

The search identified 6706 papers from the databases and 30 additional papers were obtained from other sources. After removing of duplicates and of papers that were conference papers, cross-sectional studies or animal studies, 73 papers were considered for full-text review. We excluded 42 papers based on specific reasons after full-text review and finally 28 studies were included in this report.[27–54] PRISMA flow diagram depicts the exact selection process figure 1. Across the 28 eligible studies, there were 184 543 participants (almost 31% men), with sample size ranging from 456 to 68 153. Studies were published from 1989 to 2016 and half of them (14 out of 28) took place in the USA. Studies were also conducted in England (four), Australia (three), China (two), Canada (two), Nigeria (one), the Netherlands (one) and Taiwan (one). Baseline age ranged from 14 to 109 years and follow-up time from 2 years until death (>70 years). Five studies examined men only and two studies examined women only; the majority of studies reported results for mixed populations (table 1).

Healthy ageing, and any other term used as a synonym, was defined by including various domains of information in each study. These were grouped in the following categories: survival to a specific age or during follow-up, health status (either self-reported or measured by specific questionnaires), physical performance (including information regarding mobility, disabilities and/or difficulties in activities of daily living and instrumental activities of daily living), diseases (including chronic diseases and cancer), mental health and cognition status, subjective measurements of the participants (life satisfaction, happiness and pain) and other (anthropometric measurements, personal assistance, social support) (figure 2). Most of the studies (23 out of 28) included physical performance in their definition of healthy ageing and more than half of them included information regarding diseases and mental health (18 and 16 out of 28, respectively). Survival to a specific age was also an area often found in the definition of healthy ageing, whereas health status and subjective measurements were not so often included. (Online supplementary table A1 presents the areas of information included in the definition of healthy ageing per study.)

In table 2, the analytical results of this systematic review are presented. Relevant statistics per study are provided for every group of smoking and alcohol consumption variable and for the most and least adjusted models; confounders used for the final adjustment are also provided.

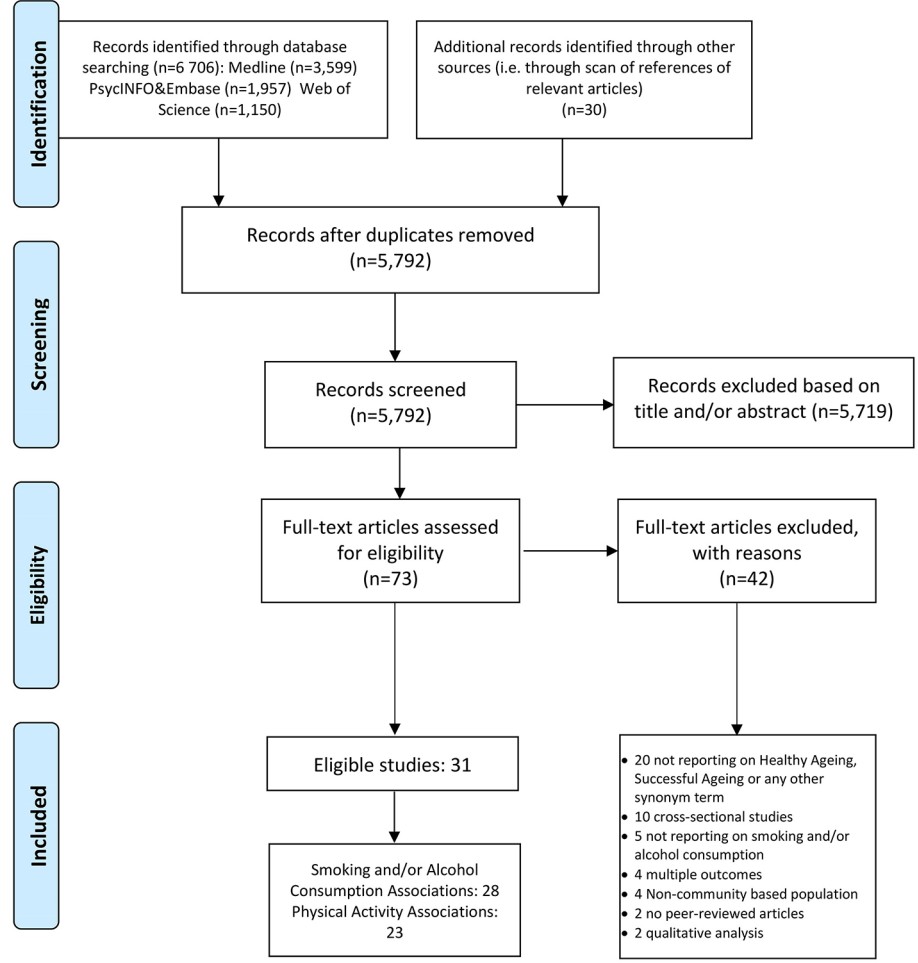

**Figure 1** Selection process.

## Quality assessment

Of the 28 studies, 1 was evaluated as having high risk of bias, 9 as moderate and 18 as having low. In aggregate, the quality of the included studies was above moderate. As it was expected, attrition and confounding were the issues that primarily contributed to moderate and high bias. Specifically, 18 out of 28 studies reported moderate or high risk of bias regarding the fact that the population lost to follow-up may be associated with key characteristics that could influence the observed relationship between outcome and factors. Similar results were also observed for the confounders' domain, where 18 out of 23 studies were characterised as having moderate risk of bias, meaning that important confounders may have not been appropriately accounted in the final model (online supplementary table A2).

## Meta-analysis

### Smoking and healthy ageing

From the 27 studies providing results for smoking, we did a meta-analysis with 18 studies. We excluded studies that provided results using (i) healthy/successful years,[30 42] (ii) risk ratio,[40] (iii) coefficients from linear mixed models or generalised estimation equation models[41 50] and (iv)

cigarette packs.[45 52 53] All studies included in the meta-analysis were of moderate and low risk of bias.

Never smokers compared with current smokers had more than double increased odds of ageing healthily (OR 2.36, 95% CI 2.03 to 2.75, $I^2$ 43.30%, 7 studies) even after adjusting for publication bias with the trim-and-fill algorithm (OR 2.11, 95% CI 1.82 to 2.45, 10 studies). Never smokers had increased ORs compared with former smokers as well (OR 1.32, 95% CI 1.23 to 1.41, $I^2$ 32.80%, five studies); no change of the combined effect estimate was observed by the trim-and-fill method. Never smokers were more likely to experience healthy ageing than current or former smokers (OR 1.29, 95% CI 1.16 to 1.43, $I^2$ 0.00%, five studies), and this finding marginally altered when we adjusted for publication bias (OR 1.27, 95% CI 1.14 to 1.40, nine studies). Never or former smokers were more likely to age in a healthy way compared with current smokers (OR 1.72, 95% CI 1.20 to 2.47, $I^2$ 87.20%, six studies); however, when we adjusted for publication bias this finding was no longer significant (OR 1.20, 95% CI 0.83 to 1.73, nine studies).

There were also two studies that reported HRs[47 48]; the pooled HR of never smokers compared with current smokers was 1.55 (95% CI 1.18 to 2.02) while the pooled

**Table 1** Baseline characteristics of the eligible studies

| Studies/authors | Country | Panel | Data collection period | Follow-up (years)[*] | Sample size | Gender | Baseline age (years) |
|---|---|---|---|---|---|---|---|
| Andrews et al[27] | Australia | Australian Longitudinal Study of Aging | 1992 | 8 | 1403 | 55% men | >70 |
| Bell et al[28] | USA | Honolulu Heart Program | 1991–1993 | Up to 21 years | 1292 | 100% men | 71–82 |
| Britton et al[29] | England | Whitehall II study | 1985–1988 | 17 | 5823 | 71% men | 35–55 |
| Burke et al[30] | USA | Cardiovascular Health Study | 1989–1990, 1992–1993 | 6.5 and 3.5 | 3342 | 39% men | >65 |
| Ford et al[31] | USA | N/A | 1993 | 2 | 602 | 33% men | >70 |
| Gu et al[32] | China | Chinese Longitudinal Healthy Longevity Survey | 2002 | 3 | 15972 | 45% men | 65–109 |
| Guralnik and Kaplan[33] | USA | Alameda County Study | 1965 | 19 | 841 | N/A | 46–70 |
| Gureje et al[34] | Nigeria | Ibadan Study of Ageing | August 2003 to November 2004 | 5.3 | 930 | 61% men | >65 |
| Hamer et al[35] | England | English Longitudinal Study of Ageing | 2002–2003 | 8 | 3454 | 42% men | 63.7 (M) |
| Hodge et al[36] | Australia | Melbourne Collaborative Cohort Study | 1990–1994 | 11.7 | 5512 | 37% men | 63 (M) |
| Hodge et al[37] | Australia | Melbourne Collaborative Cohort Study | 1990–1994 | 11.1 (WM) | 6309 | 39% men | 64.1 (WM) |
| Kaplan et al[38] | Canada | Canadian National Population Health Survey | 1994–1995 | 10 | 2432 | 44% men | 65–85 |
| LaCroix et al[39] | USA | Women's Health Initiative | 1993–1998 | 16 | 68153 | 100% women | 68.9 (WM) |
| Li et al[40] | China | Shanghai Mental Health Center | 1987 | 5 | 3024 | 43% men | 67.3 (M) |
| Liu and Su[41] | Taiwan | Taiwan Longitudinal Study on Aging | 1993 | 14 | 3155 | 56% men | 71.7 (M) |
| Newman et al[42] | USA | Cardiovascular Health Study | 1989–1990, 1992–1993 | 8 | 2932 | 39% men | 71.9 (WM) |
| Newson et al[43] | Netherlands | Rotterdam Study | 1990–1993 | 7.9 | 2008 | 34% men | 75.8 (M) |
| Pruchno and Wilson-Genderson[44] | USA | Ongoing Research on Aging in New Jersey: Bettering Opportunities for Wellness in Life | 2006–2008 | 4 | 2614 | 37% men | 60.5 (WM) |
| Reed et al[45] | USA | Honolulu Heart Program | 1965–1968 | 28 | 8006 | 100% men | 45–68 |
| Sabia et al[46] | England | Whitehall II study | 1991–1994 | >16.3 (med) | 5100 | 70% men | 51.3 (M) |
| Sarnak et al[47] | USA | Cardiovascular Health Study | 1989–1990 and 1992–1993 (African-American cohort) | 4.3 | 2140 | 38% men | 74 |
| Shields and Martel[48] | Canada | National Population Health Survey | 1994–1995 | 8 | 1309 | N/A | >65 |
| Sun et al[49] | USA | Nurses' Health Study | 1980–1986 | 20 and 16 | 13894 | 100% women | 58 (med) |
| Tampubolon[50] | England | English Longitudinal Study of Ageing | 2004 | 9 | 14765 | 46% men | 50–89 |
| Terry et al[51] | USA | Framingham Heart Study | 1948–1971 | 45 | 2531 | 44% men | 40–50 |

Continued

**Table 1** Continued

| Studies/authors | Country | Panel | Data collection period | Follow-up (years)* | Sample size | Gender | Baseline age (years) |
|---|---|---|---|---|---|---|---|
| Vaillant and Mukamal[52] | USA | Study of Adult Development at Harvard University | Circa 1940 | Until 60 or death | 724 | 100% men | Born mainly in the 1920s |
| Vaillant and Western[53] | USA | Study of Adult Development | 1940–1945 | 60 (until 70 years or death) | 456 | 100% men | 14 |
| Willcox et al[54] | USA | Honolulu Heart Program/ Honolulu Asia Aging Study | 1965–1968 | 40 | 5820 | 100% men | 54 (WM) |

*Mean years, unless otherwise specified.
M, mean; med, median; N/A, not available; WM, weighted mean.

HR of former smokers compared with never smokers was 1.30 (95% CI 1.07 to 1.59). Egger test was not significant in any analysis that we performed (p value>0.05), hence no evidence for publication bias was provided (table 3). The synthesis of the results is provided in figure 3.

### Subgroup analysis
#### Baseline age
The association of smoking with healthy ageing did not change direction or strength in the subgroup analyses. However, the beneficial association of non-smoking was higher in studies with baseline mean age <65 years. The only association that differed was that in the analysis of former or never compared with current smokers. Studies with relatively older participants showed a non-significant association (OR 1.34, 95% CI 0.65 to 2.76), whereas studies with relatively younger participants showed a statistically significant association (OR 1.90, 95% CI 1.51 to 2.40) (online supplementary figure A1).

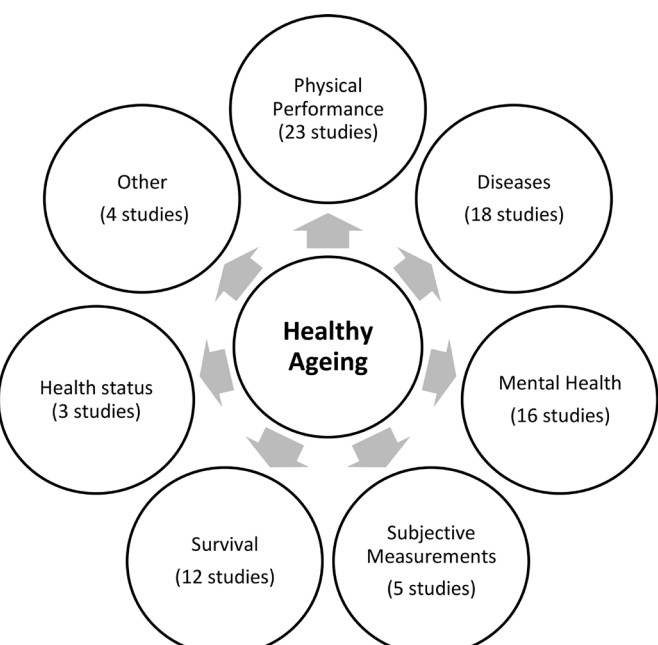

**Figure 2** Areas of information included in the definition of healthy ageing.

#### Follow-up time
We performed subgroup analyses for former or never versus current smokers and never versus former or current smokers; all other groups had follow-up time >10 years. In the group of former or never smokers compared with current smokers, the association was not significant in studies with <10 years follow-up (OR 1.41, 95% CI 0.99 to 2.01). No considerable differences were observed in the group of never compared with former or current smoking with healthy ageing (online supplementary figure A2).

### Alcohol consumption and healthy ageing
From the 21 studies that provided statistics for the association of alcohol consumption with healthy ageing, we performed meta-analyses using nine studies. We excluded studies for the following reasons: (i) the standardisation of alcohol consumption was not possible[28 38 40 44 46 48 52–54] and (ii) they provided healthy years[30] or β coefficients.[41 50]

Drinkers compared with non-drinkers had increased odds of ageing healthily (OR 1.28, 95% CI 1.08 to 1.52, $I^2$ 72.10%, five studies); the same beneficial association of alcohol was also observed when we pooled results among studies that reported findings of light consumption (<1 drinker per day) and moderate consumption (1–2 drinks per day) compared with non-drinkers: (OR 1.12, 95% CI 1.03 to 1.22, $I^2$ 0.00%, three studies); (OR 1.35, 95% CI 0.93 to 1.97, $I^2$ 71.40%, four studies). However, in the latter comparison (ie, moderate consumption compared with non-drinkers) results were not statistically significant (p value 0.112). We also pooled results of the studies reporting high (>2–4 drinks per day) consumption compared with non-drinkers; (OR 1.25, 95% CI 1.09 to 1.44, $I^2$ 0.00%, three studies). Results changed when we used the trim-and-fill algorithm only to the case of moderate to non-drinkers (OR 1.10, 95% CI 0.77 to 1.57, six studies). Egger test of bias was significant only in the analysis of light drinkers (p value 0.043) (table 3). The synthesis of the results is provided in figure 4.

**Table 2** Aggregated results of smoking and alcohol consumption to healthy ageing

| Authors | Smoking variable | Statistics and 95% CI | Alcohol variable | Statistics and 95% CI | Confounders |
|---|---|---|---|---|---|
| Andrews et al[27] | Past smoker (yes or no) | Not statistically significant (N/A) | – | – | Sociodemographic, economic |
| Bell et al[28] | Past, current, never | Unhealthy vs healthy survival; OR (95% CI) Past: 1.03 (0.78 to 1.35) Current: 1.99 (1.06 to 3.75) Never: ref | Never (0 ounces/month) Moderate to heavy (>15 ounces/ month, >1 drink per day) Mild (>0–15 ounces/ month, ≤1 drink per day) | Unhealthy vs healthy survival; OR (95% CI) Never: 1.10 (0.81 to 1.50) Moderate to heavy: 1.04 (0.73 to 1.47) Mild: ref | Sociodemographic |
| Britton et al[29] | Never, ex-smoker, current | OR (95% CI) Men: Never: 2.7 (1.8 to 4.1), not SEP adjusted: 3.2 (2.1 to 4.7) Ex-smoker: 2.5 (1.6 to 3.7), not SEP adjusted: 2.6 (1.8 to 4.0) Current smoker: ref Women: Never: 2.2 (1.3 to 3.7), not SEP adjusted: 2.4 (1.5 to 3.9) Ex-smoker: 2.2 (1.3 to 3.7), not SEP adjusted: 2.6 (1.5 to 4.5) Current smoker: ref | Alcohol consumption in the previous week (1 unit=8 g ethanol) 0 unit/week, 1–14 units/ week, 15 units/week | OR (95% CI) Men: 0: 1.2 (0.9 to 1.8), not SEP adjusted: 1.0 (0.7 to 1.4) 1–14: 1.0 (0.8 to 1.2), not SEP adjusted: 1.0 (0.8 to 1.2) 15: ref Women 0: 0.5 (0.3 to 0.9), not SEP adjusted: 0.3 (0.2 to 0.5) 1–14: 1.0 (0.7 to 1.5), not SEP adjusted: 0.8 (0.5 to 1.1) 15: ref | Sociodemographic, economic, model characteristics |
| Burke et al[30] | Former, current, never | Proportion of HY, 95% CI Model with behavioural factors only Former: 0.75, N/A Current: 0.56, N/A Never: ref Model with behavioural factors and subclinical disease factors Former: 0.78 (0.69 to 0.88) Current: 0.66 (0.56 to 0.78) Never: ref | Wine drink: yes, no | Proportion of HY, 95% CI Model with behavioural factors only Wine drink: 1.11 (0.99 to 1.24) Model with behavioural factors and subclinical disease factors Wine drink: 1.04 (0.93 to 1.17) | Sociodemographic, economic, health behaviour, diseases and physical measurements |
| Ford et al[31] | Current smoker: yes, no | OR (95% CI): No: 2.14 (1.02 to 4.48) Yes: ref | Current drinker (wine, beer or liquor): yes, no | OR (95% CI): No: 1.00 (0.59 to 1.71) | Sociodemographic, economic, health behaviour, diseases and physical measurements, attitude and social environment |

Continued

**Table 2** Continued

| Authors | Smoking variable | Statistics and 95% CI | Alcohol variable | Statistics and 95% CI | Confounders |
|---|---|---|---|---|---|
| Gu et al[32] | Current smoker: yes, no | OR (95% CI) of access to healthcare at present and in childhood on healthy survival Model I: 1.05 (0.91 to 1.21) Model II: 1.00 (0.86 to 1.16) Model III: 1.00 (0.86 to 1.16) No: ref | Current drinker: yes, no | OR (95% CI) of access to healthcare at present and in childhood on healthy survival Model I: 1.21 (1.05 to 1.39) Model II: 1.11 (0.96 to 1.28) Model III: 1.10 (0.96 to 1.27) No: ref | Model I: sociodemographic, economic Model II: sociodemographic, economic, attitude and social environment Model III: sociodemographic, economic, attitude and social environment, model characteristics |
| Guralnik and Kaplan[33] | Past-never, current | OR (95% CI) High function vs all others (dead or lower/moderate) Health practices model: past-never: 2.8 (1.6 to 4.8), current: ref Combined model: past-never: 3.0 (1.8 to 5.1), current: ref High function vs deceased Past-never: 6.1 (3.3 to 8.3), current: ref High function vs low/moderate Past-never: 2.2 (1.3 to 3.8), current: ref | None, 1–60, >60 (drinks/month) | OR (95% CI) High function vs all others (dead or lower/moderate) Health practices model: 1–60 vs none: 2.4 (1.2 to 4.8) 1–60 vs >60: 1.7 (0.6 to 5.0) Combined model: 1–60 vs none: 2.1 (1.1 to 4.1), 1–60 vs >60: 1.7 (0.6 to 5.1) High function vs deceased 1–60 vs none: 3.1 (1.5 to 6.5), 1–60 vs >60: 2.5 (0.7 to 8.2) High function vs low/moderate 1–60 vs none: 2.2 (1.1 to 4.4) 1–60 vs >60: 1.3 (0.4 to 4.0) | Health practices model: health behaviour, physical measurement Combined model: sociodemographic, economic, diseases, physical measurement, health behaviour |
| Gureje et al[34] | Ever smoking: yes, no | OR (95% CI) Total: no: 1.6 (0.71 to 3.46), yes: ref Male: no: 4.7 (1.61 to 14.52), yes: ref Female: no: 0.4 (0.22 to 1.13), yes: ref | Ever having alcohol: yes, no | OR (95% CI) Total: no: 1.0 (0.51 to 2.10), yes: ref Male: no: 0.5 (0.10 to 2.73), yes: ref Female: no: 0.3 (0.47 to 2.18), yes: ref | Sociodemographic, economic, health behaviour, diseases and physical measurements, attitude and social environment |
| Hamer et al[35] | Current, ex/non-smoker | OR (95% CI) Current: 0.66 (0.48 to 0.89) Ex/non-smokers: ref | Daily, at least weekly, rarely, never | – | Sociodemographic, economic, health behaviour |
| Hodge et al[36] | Former, current, never | OR (95% CI) Former: 0.72 (0.61 to 0.84) Current: 0.45 (0.32 to 0.65) Never: ref | Alcohol intake (g/day) | OR (95% CI) 1–20: 1.06 (0.90 to 1.24) 21–40: 1.16 (0.93 to 1.46) 41–60: 1.02 (0.72 to 1.45) 60+: 1.25 (0.79 to 1.97) None: ref | Sociodemographic, economic, health behaviour, diseases and physical measurements, attitude and social environment |

Continued

**Table 2** Continued

| Authors | Smoking variable | Statistics and 95% CI | Alcohol variable | Statistics and 95% CI | Confounders |
|---|---|---|---|---|---|
| Hodge et al[37] | Former, current, never | OR (95% CI) Model without BMI and WHR Former: 0.67 (0.59 to 0.77) Current: 0.39 (0.30 to 0.50) Never: ref Model with BMI and WHR Former: 0.69 (0.60 to 0.79) Current: 0.38 (0.29 to 0.48) Never: ref | Alcohol intake (g/day) | OR (95% CI) Model without BMI and WHR 1–20: 1.19 (1.03 to 1.36) 21–40: 1.37 (1.11 to 1.67) 41–60: 1.27 (0.94 to 1.74) 60+: 1.17 (0.78 to 1.76) None: Ref Model with BMI and WHR 1 to 20: 1.17, (1.02 to 1.34) 21 to 40: 1.33, (1.09 to 1.63) 41 to 60: 1.29, (0.95 to 1.76) 60: 1.20, (0.80 to 1.80) None: Ref | Sociodemographic, economic, health behaviour, diseases and physical measurements, model characteristics |
| Kaplan et al[38] | Smoker, never smoker | OR (95% CI) Thrivers vs non-thrivers: smoker: 1.89 (1.10 to 3.23), never smoker: ref Thrivers vs deceased: Smoker: 4.35 (2.44 to 7.69), never smoker: ref | Moderate (1–14 drinks/week), none or heavy | OR (95% CI) Thrivers vs non-thrivers, moderate: 1.78 (1.07 to 2.95), none-heavy: ref Thrivers vs deceased, moderate: 2.22 (1.25 to 3.95), none-heavy: ref | Sociodemographic, economic, attitude and social environment, health behaviour, diseases and physical measurements |
| LaCroix et al[39] | Current, past, never | OR (95%) Crude model, veterans Current: 0.35 (0.22 to 0.57), past: 0.85 (0.71 to 1.02), never: ref Adjusted model, veterans Current: 0.31 (0.19 to 0.52), past: 0.80 (0.66 to 0.97), never: ref Crude model, non-veterans Current: 0.49 (0.45 to 0.54), past: 0.83 (0.80 to 0.86), never: ref Adjusted model, non-veterans Current: 0.50 (0.46 to 0.55), past: 0.77 (0.74 to 0.80), never: ref Model for veterans only Crude: current: 0.35 (0.22 to 0.57), past: 0.85 (0.71 to 1.02), never: ref Adjusted: current: 0.31 (0.18 to 0.52), past: 0.79 (0.65 to 0.95), never: ref | Drinker, non-drinker | OR (95%) Crude model, veterans Non-drinker: 0.59 (0.48 to 0.73), drinker: ref Adjusted model, veterans Non-drinker: 0.64 (0.51 to 0.81), drinker: ref Crude model, non-veterans Non-drinker: 0.67 (0.64 to 0.69), drinker: ref Adjusted model, non-veterans Non-drinker: 0.71 (0.68 to 0.74), drinker: ref Model for veterans only: Crude: non-drinker: 0.59 (0.48 to 0.73), drinker: ref Adjusted: non-drinker: 0.62 (0.49 to 0.78), drinker: ref | Crude: age adjusted: adjusted: sociodemographic, economic, model characteristics, health behaviour, diseases and physical measurements |
| Li et al[40] | Current, former, never | RR (95% CI) Current: 1.11 (1.01 to 1.21) Former: 0.8, (0.74 to 1.03) Never: 0.96 (0.88 to 1.05) | Current, former, non-drinker | RR (95% CI) Current: 1.16 (1.07 to 1.27) Former: 0.87 (0.68 to 1.12) Never: 0.89 (0.81 to 0.96) | Sociodemographic, health behaviour |
| Liu and Su[41] | Smoking (yes/no) | β=0.042, SE=0.03, p value: 0.163 No: ref | Alcohol (yes/no) | β=0.201, SE: 0.03, p value<0.001 No: ref | Sociodemographic, health behaviour, attitude and social environment |

Continued

**Table 2** Continued

| Authors | Smoking variable | Statistics and 95% CI | Alcohol variable | Statistics and 95% CI | Confounders |
|---|---|---|---|---|---|
| Newman et al[42] | Self-reported pack-years: None <10 >10–20 >20–40 >40 | Proportion of SY (95% CI) Current smoker: 0.77 (0.65 to 0.90) (multiplication factor) <10: 0.84 (0.73 to 0.98) >10–20: 0.90 (0.76 to 1.07) >20–40: 0.99 (0.85 to 1.15) >40: 0.82 (0.71 to 0.95) None: ref | – | – | Sociodemographic, economic, health behaviour, diseases and physical measurements |
| Newson et al[43] | Never, former or current | Survival morbidity free vs non-survival, OR (95% CI) Age and sex-adjusted model: 1.38 (1.07 to 1.77) Full model: 1.26 (0.97 to 1.65) | Average consumption (g)/day | Not statistically significant | Sociodemographic, economic, health behaviour, diseases and physical measurements, attitude and social environment |
| Pruchno and Wilson-Genderson[44] | Current smoker (yes, no) | Successful: ref Unsuccessful: b=0.55, SD=0.26, 95% CI 1.03 to 2.93, exp(b)=1.74 Subjective only: b=0.32, SD=0.25, 95% CI 0.83 to 2.27, exp(b)=1.38 Objective only: b=0.24, SD=0.23, 95% CI 0.81 to 1.99, exp(b)=1.27 | Days per week where have at least one drink of alcohol (0–7) | Successful: ref Unsuccessful: b=−0.09, SD=0.08, 95% CI 0.83 to 1.00, exp(b)=0.91 Subjective only: b=−0.02, SD=0.04, 95% CI 0.90 to 1.06, exp(b)=0.98 Objective only: b=0.01, SD=0.03, 95% CI 0.94 to 1.08, exp(b)=1.01 | Sociodemographic, economic, health behaviour, attitude and social environment |
| Reed et al[45] | Cigarette pack-years = (usual number of cigarettes/day) * (number of years) | OR (95% CI) Healthy vs illness with impairment: 0.52 (0.39 to 0.70) Healthy vs illness without impairment: 0.46 (0.33 to 0.64) Healthy vs impairment without illness: Survival ratio, 95% CI: 0.72 (0.65 to 0.79) | Monthly intake of beer, wine and liquor (ml of ethanol/day) | – | Sociodemographic, health behaviour, diseases and physical measurement, model characteristics |
| Sabia et al[46] | Current or former smoker, never | OR (95% CI) Never smoker: 1.29 (1.11 to 1.49) Former or current smoker: ref | Abstinence: no alcohol in the last week Moderate: 1–14 units/week women; 1–21 units/week men Heavy: ≥15 units/week women; ≥21 units/week men | OR (95% CI) Moderate: 1.31 (1.12 to 1.53) No moderate: ref | Sociodemographic, economic, health behaviour |
| Sarnak et al[47] | Current, former, never | HR (95% CI) Current: 1.73 (0.79 to 1.80) Former: 1.21 (0.92 to 1.58) Never: ref | – | – | Sociodemographic, diseases and physical measurements |

Continued

**Table 2** Continued

| Authors | Smoking variable | Statistics and 95% CI | Alcohol variable | Statistics and 95% CI | Confounders |
|---|---|---|---|---|---|
| Shields and Martel[48] | Current daily smokers, had quit daily smoking within the past 15years, had quit at least 15years ago or had never smoked every day | HR (95% CI) Current: 0.7 (0.5 to 1.0) Quit during the past 15years: 0.7 (0.5 to 0.9) Never smoked/quit for 15+years: ref | Weekly/occasional drinker, non-drinker | HR (95% CI) Weekly/occasional drinker: 1.4 (1.1 to 1.8) Non-drinker: ref | Sociodemographic, economic, health behaviour, attitude and social environment |
| Sun et al[49] | – | – | Alcohol intake levels (g/day) (i) 0 (ii) ≤5.0 (iii) 5.1–15.0 (iv) 15.1–30.0 vs 30.1–45.0 | Fully adjusted OR (95% CI) (i) 0: ref (ii) ≤5.0: 1.11 (0.96 to 1.29) (iii) 5.1–15.0: 1.19 (1.01 to 1.40) (iv) 15.1–30.0: 1.28 (1.03 to 1.58) vs 30.1–45.0: 1.24 (0.87 to 1.76) Age-adjusted OR (95% CI) (i) 0: ref (ii) ≤5.0: 1.12 (0.97 to 1.29) (iii) 5.1–15.0: 1.22 (1.05 to 1.43) (iv) 15.1–30.0: 1.26 (1.03 to 1.53) (v) 30.1–45.0: 1.05 (0.75 to 1.46) | Sociodemographic, economic, health behaviour, diseases and physical measurements |
| Tampubolon[50] | Current, no current | Baseline model, annual rate of phenotypic decline: b=−0.679, 95% CI −0.852 to 0.507 Gender interaction model, annual rate of phenotypic decline: b=−0.685, 95% CI −0.858 to 0.513 | Drink daily Less than daily: ref | Baseline model, annual rate of phenotypic decline: b=0.414, 95% CI 0.309 to 0.519 Gender interaction model, annual rate of phenotypic decline: b=0.412, 95% CI 0.307 to 0.517 | Baseline model: sociodemographic, economic, health behaviour, diseases and physical measurements |
| Terry et al[51] | Present (smoked within the year before any baseline examination), absent | OR (95% CI) 0.51 (0.41 to 0.63) | – | – | Sociodemographic, economic, health behaviour, diseases and physical measurement |
| Vaillant and Mukamal[52] | Smoking<30 pack-years (from age 15–50) Smoking≥30 pack-years (from age 15–50) | OR (95% CI) Happy-well men vs Sad-sick or prematurely dead Smoking<30 pack-years: college men at age 75–80, 4.81 (0.84 to 27.7) Smoking<30 pack-years: Core-city men at age 65–70, 4.56 (2.29 to 9.11) Smoking in≥30 pack-years: ref | Alcohol abuse absent (yes, no) | OR (95% CI) Happy-well men vs sad-sick or prematurely dead No alcohol abuse: college men at age 75–80 No alcohol abuse: core-city men at age 65–70, 1.11 (0.53 to 2.35) | Sociodemographic, diseases and physical measurement, health behaviour, attitude and social environment |

**Table 2** Continued

| Authors | Smoking variable | Statistics and 95% CI | Alcohol variable | Statistics and 95% CI | Confounders |
|---|---|---|---|---|---|
| Vaillant and Western[53] | 0–30 pack-years, ≥30 pack-years | OR (95% CI) Happy-well vs sad-sick and prematurely dead Univariate model: smoking<30 pack-years: 5.08 (2.85 to 9.0) Multivariate model: smoking<30 pack-years: 5.00 (2.38 to 10.5) | Alcohol abuse absent (yes, no) | OR (95% CI) Happy-well vs sad-sick and prematurely dead Univariate model: no alcohol abuse: 2.64 (1.45 to 4.81) Multivariate model: no alcohol abuse: 0.94 (0.72 to 2.05) | Univariate: diseases and physical measurements Multivariate: sociodemographic, economic, health behaviour, diseases and physical measurement, attitude and social environment |
| Willcox et al[54] | Ever smoker, never smoker | OR (95% CI) Usual survival vs exceptional survival Age adjusted: 1.27 (1.06 to 1.53) Fully adjusted: 1.23 (1.01 to 1.50) Never smokers: ref | 3 or more drinks/day, <3 drinks/day | OR (95% CI) Usual survival vs exceptional survival Age adjusted: 1.84 (1.29 to 2.62) Fully adjusted: 1.61 (1.11 to 2.34) Less than 3 drinks/day: ref | Sociodemographic |

OR, odds ratio; CI, confidence interval; SEP, socioeconomic position; HY, healthy years; WHR, waist hip ratio; RR, risk ratio; SY, successful years; BMI, body mass index; N/A, not available.

### Subgroup analysis
#### Baseline age
We could not perform subgroup analyses for the associations of alcohol consumption with healthy ageing as studies were homogeneous.

#### Follow-up time
We performed subgroup analysis for drinkers versus non-drinkers; all other groups were homogeneous (ie, >10 years). The association of alcohol consumption with healthy ageing was significantly positive in studies with follow-up time >10 years (OR 1.41, 95% CI 1.36 to 1.48), whereas studies with follow-up time <10 years showed a non-significant beneficial association (OR 1.09, 95% CI 0.95 to 1.24) (online supplementary figure A3).

### DISCUSSION
Based on our knowledge, this is the first systematic review to examine the association of smoking and alcohol consumption with healthy ageing, which included a meta-analysis to produce a pooled effect estimate and also examined eligible studies for publication bias. In total, 23 out of 27 studies reported a positive association between never or former smoking and healthy ageing and four reported a non-significant relationship. With regards to alcohol consumption, results were mixed but a positive association of drinking with healthy ageing was reported in 12 out of 22 studies. Nine studies did not find an association and one study reported a negative one.

The current systematic review adds to the plethora of evidence that smoking is associated with worse health outcomes in older age. In fact, the only study that reported a positive association between current smoking and healthy ageing was that of Li et al[40] where smokers exhibit increased risk ratios of healthy survival (RR 1.11, 95% CI 1.01 to 1.21); however, the authors recommend to treat this finding with caution as the population-attributable percentage is very low after 5 years of follow-up (2.55%). In aggregate, being a non-smoker or a former smoker increases considerably the odds of ageing healthily. From our meta-analysis, we found that never smokers have more than double the odds of experiencing healthy ageing (OR 2.36, 95% CI 2.03 to 2.75) compared with current smokers. Furthermore, never smokers are also more likely to age in a healthy way by >30% (OR 1.32, 95% CI 1.23 to 1.41) compared with former smokers. These findings indicate that smoking cessation is always beneficial and comes in accordance with a 50-year study on doctors' mortality in relation with smoking. In this study, it was revealed that smoking cessation at age 60, 50, 40 or 30 years could increase life expectancy by 3, 6, 9 or 10 years, respectively.[55] Our findings also confirm previous research examining the beneficial effect of non-smoking on successful ageing.[56] In addition, we found that never/former smokers compared with current smokers have an increased likelihood of healthy ageing (OR 1.72, 95% CI 1.20 to 2.47), which becomes

**Table 3** Results of meta-analysis: healthy ageing compared with smoking and alcohol consumption

| Analysis | Studies (n) | ORs and 95% CI | P values | $I^2$ (%) | Egger bias; P values | Trim-and-fill ORs and 95% CI | Filled studies (n) |
|---|---|---|---|---|---|---|---|
| Never vs current smokers | 7* | 2.36 (2.03 to 2.75) | <0.001 | 43.3 | 1.42; 0.060 | 2.11 (1.82 to 2.45) | 10 |
| Never vs former smokers | 5† | 1.32 (1.23 to 1.41) | <0.001 | 32.8 | −0.09; 0.935 | Unchanged | – |
| Past/never vs current | 6‡ | 1.72 (1.20 to 2.47) | 0.003 | 87.2 | 3.55; 0.142 | 1.20 (0.83 to 1.73) | 9 |
| Never vs past/current | 5§ | 1.29 (1.16 to 1.43) | <0.001 | 0.0 | 1.08; 0.157 | 1.27 (1.14 to 1.40) | 9 |
| Drinkers vs non-drinkers | 5¶ | 1.28 (1.08 to 1.52) | 0.004 | 72.1 | −1.25; 0.350 | Unchanged | – |
| Light vs non-drinkers | 3** | 1.12 (1.03 to 1.22) | 0.010 | 0.0 | −8.17; 0.043 | Unchanged | – |
| Moderate vs non-drinkers | 4†† | 1.35 (0.93 to 1.97) | 0.112 | 71.4 | 1.77; 0.549 | 1.10 (0.77 to 1.57) | 6 |
| High to non-drinkers | 3‡‡ | 1.25 (1.09 to 1.44) | 0.002 | 0.0 | −0.63; 0.828 | Unchanged | – |

*Bell et al[28]; Britton et al (males)[29]; Britton et al (females)[29]; Hodge et al[36]; Hodge et al[37]; LaCroix et al (veterans)[39]; LaCroix et al (non-veterans).[39]
†Bell et al[28]; Hodge et al[36]; Hodge et al[37]; LaCroix et al (veterans)[39]; LaCroix et al (non-veterans).[39]
‡Ford et al[31]; Gu et al[32]; Guralnik and Kaplan [33]; Hamer et al[35]; Pruchno and Wilson-Genderson[44]; Terry et al.[51]
§Kaplan et al[38]; Newson et al[43]; Gureje et al[34]; Sabia et al[46]; Willcox et al.[54]
¶Ford et al[31]; Gu et al[32]; Gureje et al[34]; LaCroix et al (veterans)[39]; LaCroix et al (non-veterans).[39]
**Hodge et al[36]; Hodge et al[37]; Sun et al.[49]
††Britton et al (males)[29]; Britton et al (females)[29]; Guralnik and Kaplan[33]; Sun et al.[49]
‡‡Hodge et al[36]; Hodge et al[37]; Sun et al.[49]

non-significant when we adjust for publication bias (OR 1.20, 95% CI 0.83 to 1.73). This comes in agreement with the finding of a study examining the transitions within smoking categories and successful ageing, where former compared with current smokers cannot predict successful ageing.[57]

Positive associations between drinking and healthy ageing refer to limited alcohol consumptions; for

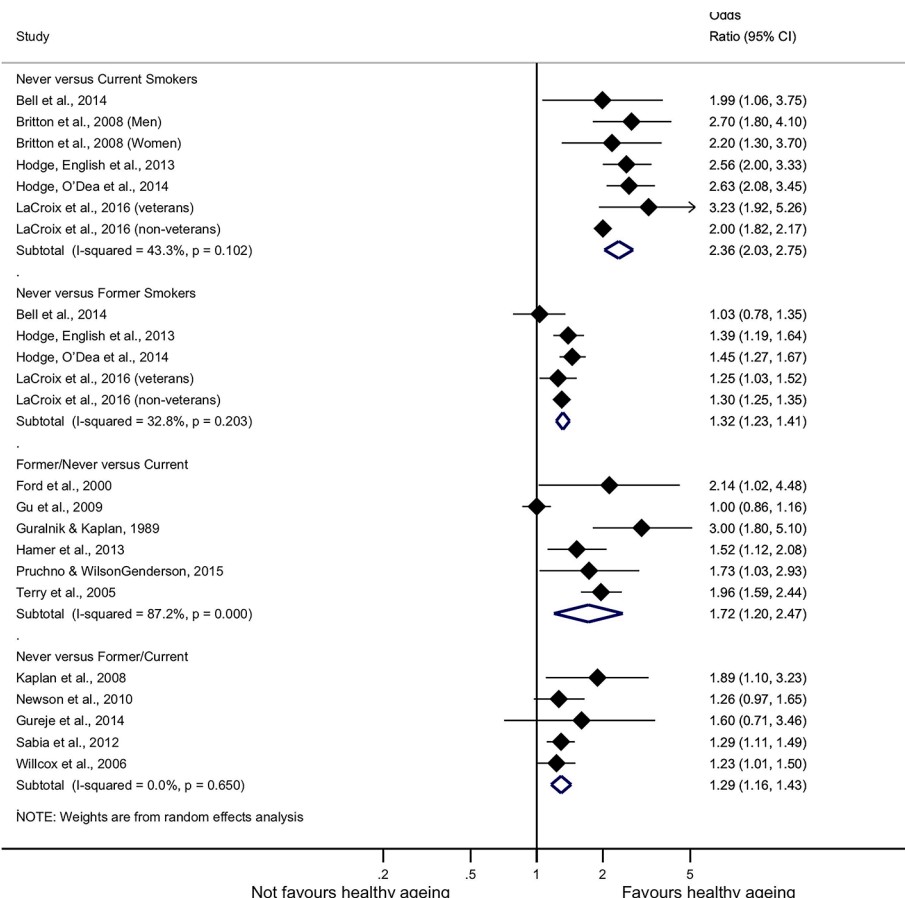

**Figure 3** Smoking and healthy ageing forest plot.

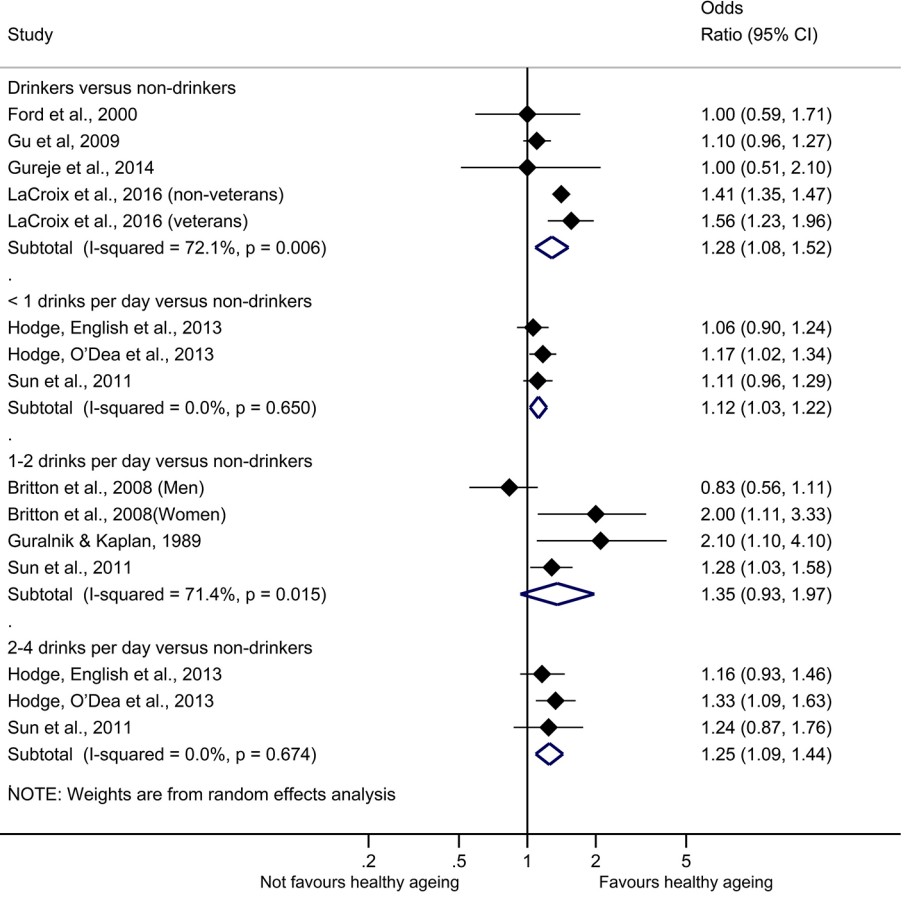

**Figure 4**  Alcohol and healthy ageing forest plot.

example, 1–40 g per day,[37] 1–14 drinks per week,[38] 1–14 units per week for women and 1–21 units per week for men.[46] The one study reporting a negative association is that of Willcox *et al*[54]; however, this finding should be interpreted with caution as drinkers with more than three drinks per day are compared with non-drinkers only and there is no comparison among light or moderate drinkers and non-drinkers. From our meta-analysis, we concluded that compared with never drinkers reasonable alcohol consumption is beneficial to healthy ageing; pooled OR for drinkers 1.28 (95% CI 1.08 to 1.52), light drinkers 1.12 (95% CI 1.03 to 1.22), moderate drinkers 1.35 (95% CI 0.93 to 1.97) and high drinkers 1.25 (95% CI 1.09 to 1.44). Nevertheless, associations are marginally statistically significant, while for the moderate category the pooled effect estimate is non-significant, so extra caution is needed before making a final conclusion.

Several studies of the associations of alcohol consumption with healthy ageing were excluded from our meta-analyses; among those there were seven studies that reported positive association between alcohol consumption and healthy ageing,[38 40 41 44 46 48 50] four studies reported non-significant association[28 30 52 53] and one study reported a negative association.[54] As a consequence, we expect the positive effect of light to moderate alcohol drinking would be reinforced if we included these studies in our meta-analyses. This finding comes in accordance

with the fact that light to moderate alcohol consumption (≤1 drink per day) is also associated with a reduced risk of multiple cardiovascular outcomes,[11] better cognition and well-being,[58] and a reduced risk of substantial functional health decline.[59]

However, relating moderate alcohol consumption with health benefits should not come without question since there are many other issues to consider. Our results could be biased by confounders that have not been taken into account, such as health status or former and occasional drinkers to be counted as non-drinkers.[60] For instance, in the subgroup of drinkers the positive effect comes mainly from the study of LaCroix *et al*[39]; however, in that study former drinkers were also assumed as non-drinkers.

In addition, since typical drink sizes vary per country as well as the recommended daily or weekly amounts,[61] we cannot be sure that participants have been assigned to the correct drinking category. Our results should also be questioned regarding the fact that alcohol consumption was only assessed at baseline (exception to this is the study of Sun *et al*)[49]; during the baseline assessment some cohorts could have had middle-aged participants drinking up to four drinks per day and that could be considered a moderate consumption. However, older people do not have the same tolerance to alcohol[62] and for them moderate consumption should consist of up to two drinks per day. Likewise, it should have

been considered that older adults tend to consume less alcohol than middle-aged groups[63] and hence baseline assessments perhaps are not representative of alcohol consumption in an older age. Nevertheless, more research is required to establish or reject the beneficial effects of limited alcohol consumption. In an extended systematic review and meta-analysis of alcohol impact on mortality risk, the benefits of low-volume drinkers largely disappeared once studies were controlled for several design and methodological issues.[64] Mendelian randomisation studies, helping to examine causal relationships in observational studies by limiting confounding and reverse causation risk of bias,[65] may also contribute to elucidating if this beneficial effect of light alcohol consumption with healthy ageing is a true biological effect or not. For instance, a recent study of this type found no causal association between alcohol consumption and Alzheimer's disease.[66]

### Strengths and limitations

The fact that this review was independently double screened, taking into account previous systematic reviews in the field and the reference lists of included papers, allows a good amount of confidence that all relevant studies were included. Regarding the quality assessment of the studies, limited disagreement among the six different domains per study was reached between the two reviewers, who independently assessed them, concluding that the quality assessment tool was reliable and did not allow great amount of misjudgement. Attrition rate and not allowing important confounders to be included in the final models were important factors contributing to the quality of the considered studies. Hence future research should consider these domains more thoroughly.

However, the following limitations have to be taken into account. Smoking and alcohol consumption were not measured or/and defined in a consistent way among the different cohorts; hence it was not possible to create a pooled effect size estimate across all eligible studies. In addition, healthy ageing definition was a source of heterogeneity since each study provided its own definition. Nevertheless, it is noteworthy that all studies assumed survival in the follow-up, even if the latter was not explicitly mentioned in the healthy ageing definition, and that many had definitions that included areas of physical performance, in agreement with the Depp and Jeste review on successful ageing.[56] In addition, most of the studies used a categorical definition of smoking, making impossible the assessment of abstinence time or pack-years on healthy ageing. Alcohol measurement and drinking categorisation were also another part of the underlying heterogeneity. Similar comments have been reported in other systematic reviews examining alcohol consumption.[64 67] Furthermore, self-reported questionnaires and the fact that each study was adjusted by using a different set of covariates, different follow-up time and attrition rate, and the different definition of healthy ageing increase the heterogeneity and the

likelihood of bias. Especially for alcohol, research has shown that self-reported consumption is mostly underestimated.[68]

Due to the limited number of studies included in our meta-analyses, we did not assess publication bias by funnel plots; instead we used Egger test, and even though it was statistically non-significant in the majority of cases, we applied the trim-and-fill method to examine the robustness of our findings. We used random-effect meta-analysis to consider the heterogeneity of the studies, even though $I^2$ was not considerable in all analyses (table 3). We did not assess heterogeneity with the Cochran's Q statistic due to the quite limited number of studies; performing meta-regression analysis and assessing the confounding effect of covariates (ie, age, gender, follow-up time) was not possible either. However, we did evaluate the robustness of our findings by also implementing the Paule and Mandel (PM) estimator to assess between study variability and estimate our pooled results as simulations have shown that the PM estimator is less biased and more efficient than other alternatives.[69] We did not observe any considerable fluctuations in our results (online supplementary table A3).

We assessed the effect of different baseline age and follow-up time to the pooled estimates by performing subgroup analyses; pooled effect estimates were not severely affected in direction or in strength. The more positive effect of non-smoking to studies with relatively younger cohorts comes in accordance with other studies examining smoking and adverse health outcomes in old populations.[70 71] When studies incorporate an older sample, this is biased in favour of people who smoke and survive compared with smokers who die (survival bias). Subgroup analyses showed that in studies with >65-year-old participants or follow-up time <10 years, the beneficial association of former/never compared with current smokers with healthy ageing is not statistically significant; result which was also indicated by the trim-and-fill analysis. Follow-up subgroup analysis confirmed that the beneficial effect of drinking compared with non-drinkers comes from the study of LaCroix et al[39] emphasising that this result should be interpreted with caution.

From our review, it is evident that the majority of the studies have been implemented in high-income countries (24 out of 28 studies). Thus, in accordance with a previous study,[72] our findings revealed the limited research on ageing in low-and-middle-income countries (LMIC), even though by 2050 80% of people aged ≥60 years will live there.[73] This is of high importance considering that alcohol use increases in LMICs[74] and that nowadays 80% of the more than one billion smokers live there.[75] Estimations also indicate that by 2030 four out of five smoking-related deaths will occur there, highlighting even more the heavy economic burden of smoking in these countries.[76]

## CONCLUSIONS

In conclusion, smoking abstinence and smoking cessation are positively associated with healthy ageing. As it takes >20 years for most smoking-related diseases to develop, the best practice would be to enforce smoking prevention policies, such as marketing bans and high taxation, and reduce smoking uptake among younger cohorts.[1] A positive relationship between limited alcohol consumption and healthy ageing could also be argued but more research is needed. From our research it becomes evident that study designs should be comparable to conclude with more confidence the generalisability of our findings. This could be achieved by adopting similar measurements of smoking and alcohol behaviour and by implementing a unanimous metric of healthy ageing. Finally, further research should focus on the examination of ageing in LMICs.

**Acknowledgements** The authors thank Chih-Cheng Chang for his contribution to the translation of the non-English study. The authors thank ATHLOS (Ageing Trajectories of Health: Longitudinal Opportunities and Synergies) project; grant agreement number 635316.

**Contributors** CD designed the study, carried out the literature review, data extraction, statistical analysis, data interpretation, article preparation, article review and correspondence. BS contributed to the data interpretation and article review. CK contributed to the literature review and data extraction. AK contributed to article preparation and article review. MP contributed to the study design, article preparation and article review. AMP contributed to the study design, data interpretation, article preparation and article review. All authors contributed to the final report and approved the final version.

**Funding** This project falls under the ATHLOS (Ageing Trajectories of Health: Longitudinal Opportunities and Synergies) project, funded by the European Union's Horizon 2020 Research and Innovation Programme under grant agreement number 635316.

**Disclaimer** The views expressed are those of the authors and not necessarily those of NIHR, the NHS or the UK Government Department of Health. The sponsor had no participation in the study design, data extraction, data interpretation or writing of this study.

**Competing interests** AMP was supported by the MRC MR/K021907/1. AK was supported by a grant funded by the National Institute for Health Research (NIHR) Biomedical Research Centre at South London and Maudsley National Health Service (NHS) Foundation Trust and Kings' College London where most of this work was conducted. AK is now formally based at the Liverpool School of Tropical Medicine.

**Patient consent** Not required.

**Provenance and peer review** Not commissioned; externally peer reviewed.

**Data sharing statement** Statistical code is available on request to the corresponding author; no other additional data are available.

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
