## [Reviewer comments · BMJ Open]

ARTICLE DETAILS

TITLE (PROVISIONAL)	Associations of Smoking and Alcohol Consumption with Healthy Ageing: A Systematic Review and Meta-analysis of longitudinal studies
AUTHORS	Daskalopoulou, Christina; Stubbs, B; Kralj, Carolina; Koukounari, Artemis; Prince, Martin; Prina, A. Matthew

VERSION 1 – REVIEW

REVIEWER	Sara Hägg Karolinska Institutet, Sweden
REVIEW RETURNED	29-Sep-2017

GENERAL COMMENTS	RE: Associations of Smoking and Alcohol Consumption with Healthy Aging – Daskalopoulou et al. The authors have made a thorough search for publications relating smoking and alcohol consumption to healthy aging. They provide all information needed in order to follow their search strategy and also make all necessary additional statistical tests where applicable. They conclude that non-smokers and also former smokers have a healthier aging trajectory compared to smokers, and that non-smokers are healthier than former smokers. For alcohol consumption, the association is not as easy to explain. The trend though is that drinkers do have a better health profile than non-drinkers. Some comments: • Healthy aging is not defined properly. What is meant by healthy aging? Is it survival, measures related to physical function or biomarkers of aging? This has to be clearly stated and should be listed for all studies. If different measures are combined, how is this done?• The association between smoking status and healthy aging follows the expected dose-response direction of effect with increased smoking giving a worse health outcome. This conclusion may be supported by many studies and also in terms of causality in Mendelian Randomization studies. Perhaps worth mentioning.• The association between alcohol consumption and healthy aging is however more difficult to interpret and does not follow the expected direction of effect. Why would drinkers in all categories be healthier than non-drinkers? This does not make any sense. There are plenty of studies showing this observational association but it is also likely influenced by many confounding factors and reverse causation. In light of this it is extremely important to try to disentangle the relationship better by perhaps adjusting for confounding factors and also look into causal studies using the Mendelian Randomization approach. There is an excellent paper
--

	now in the bioRxiv explaining all possible confounding factors and doing a Mendelian Randomization on alcohol consumption and Alzheimer's Disease which is indeed showing a null result; that no such association exist (doi: https://doi.org/10.1101/190165). The current study would benefit from presenting adjusted models and also explain the result better by providing information on possible confounding factors driving these results because they are not likely to be true.
--	--

REVIEWER	Julie Byles University of Newcastle, Australia
REVIEW RETURNED	17-Oct-2017

GENERAL COMMENTS	An interesting topic of importance - to assess the association between common and significant health risks (smoking and alcohol consumption) and healthy ageing. Followed appropriate protocols for systematic reviews including PRSIMA, MOOSE, and processes for data extraction and quality assessment with independent raters. The review was registered in PROSPERO. 73 papers were identified and 28 were included in the review. Most studies were undertaken in USA. The definition of the risk factors was complicated by:  1) A categorical definition of smoking status, without regard to pack years or time since quitting – this should be discussed. 2) Different measures of alcohol consumption across studies – this has been discussed. 3) Inability to consider ex-drinkers (sick abstainers) – this has been discussed. The definition of healthy ageing was heterogenous and included:  1) Survival, 12/28 2) Health status 3/28 3) Physical performance 23/28 studies 4) Diseases 18/28 5) Mental health/ Cognition 16/28 6) Life satisfaction 5/28 It is not clear how these very different outcomes were operationalised as “healthy ageing” in the review. It would also be useful to see if effects are different using different operational definitions. The review was also limited by high/moderate bias in the primary studies, particularly due to attrition and confounding. This has been discussed. 27 studies assessed effects of smoking and 18 were included in a meta-analysis. Studies were excluded due to differences in outcome type, or exposure classification. The results of the meta-analysis show a 2.36 increase odds of healthy ageing (variously defined) for non-smokers compared to current smokers. 21 studies assessed effects of alcohol and 9 were included in a
--

	meta-analysis. The results show drinkers had higher odds of healthy ageing compared to non-drinkers, although the association for moderate drinkers was not statistically significant. The analyses were not disaggregated to show the separate effects of smoking or alcohol on men and women. These results are consistent with most of the previous literature and discourse on the effects of smoking and alcohol and the findings from previous studies. No assessment of population attributable risk for these factors and (un) healthy ageing has been provided (considering prevalence, and changing prevalence, of exposure). The potential value of this paper is in summarizing these risks and discussing the interpretation. It is helpful to contrast the summary findings with outliers, as for example the one study finding the negative association between alcohol and healthy ageing. This sort of examination could be drawn out more with implications for both methods, and for understanding differential impacts (on different outcomes) or the effects of dose. The fact that most studies were conducted in USA and higher income countries should also be discussed. Smoking and alcohol consumption are major issues for countries with emerging economies which are also rapidly ageing. The conclusions are not strong, and do not reflect the value of the paper. They mainly focus on the lack of consensus in the measures rather than the significance of the findings and how these might be used to promote healthy ageing.
--	---

REVIEWER	Katy Tobin Trinity College Dublin, Ireland
REVIEW RETURNED	06-Nov-2017

GENERAL COMMENTS	This work aims to combine the results published in longitudinal studies regarding alcohol use and smoking and their effects on healthy ageing. The paper is well written and you have clearly highlighted the difficulties of gathering data on alcohol use and categorising these data, as well as highlighting the need for similar research in low and middle income countries. I have a few suggestions for minor amendments: Introduction, line 12: Consider changing "elderly" to "older". Introduction, line 32: Sensitivity to the effects of alcohol, rather than "sensitivity in alcohol" Introduction, line 36: Reference 8 is for a book. Please add the specific page number that the reference applies to. Do you mean that the consumption of some amount of alcohol has a different effect depending on age, or that consuming some amount at a young age can create a problem in later years? I expect you mean the former, but the language used is unclear. Introduction, line 39: "older people with drinking problems often do not present to services due to stigma associated with the condition". Consider changing to "older people who drink harmfully often do not present to services due to stigma associated with harmful drinking".
--

	In my own writing I try to be careful of defining people by an illness that they suffer from, or a behaviour that they demonstrate. Therefore, rather than label a group as “drinkers” or “smokers”, I would use the terms “people who drink” and “people who smoke”. You might consider this in your work. Conclusion, line 48: “A positive relationship between limited consumptions of drinkin
--	---

REVIEWER	Abdelmonem A. Afifi Professor of Biostatistics Fielding School of Public Health UCLA Los Angeles, CA, USA
REVIEW RETURNED	01-Dec-2017

GENERAL COMMENTS	General comment The purpose of this study is to conduct a systematic review and meta-analysis of longitudinal studies to synthesize the associations of smoking and alcohol consumption with healthy ageing (HA). To this end, the authors searched major data bases and identified 28 studies for inclusion in their analysis. The statistical analysis produced a pooled effect estimate of various categories of these two risk factors. The authors performed a random effects meta-analysis, and checked heterogeneity and publication bias. They also adjusted for potential publication bias. The authors seem to have used the methods and interpreted the results correctly. Specific comment The authors use the term “late survival” without defining it. Please define.
---

REVIEWER	Andrea Benedetti McGill University Canada
REVIEW RETURNED	04-Dec-2017

GENERAL COMMENTS	This is a very interesting paper that tries to meta analyze the associations between smoking and healthy ageing and alcohol and healthy ageing. I have several concerns, mostly about the heterogeneity across studies. Confounding related concerns:  1. Studies that were addressing the association between another exposure and healthy ageing but included smoking and alcohol as covariates were included in this meta analysis – were relevant confounders accounted for in these studies? 2. More information on whether it is believed that the studies contributing results for alcohol were adequately adjusted for smoking and other important confounders is necessary. Similarly for those reporting a smoking-outcome association. The list of variables provided makes it difficult to assess if the right variables were adjusted for. I would like to know the proportion believed to be adequately adjusted. Perhaps stratifying if there are many that are not adequately adjusted would be appropriate.
---

	Outcome related concerns: 3. The outcome definition is difficult for me. What is “healthy ageing”? You exclude studies where this was self report – how was it assessed in the other studies? Similarly around the idea that if this was reported as multiple outcomes. More detail is needed here. 4. In the results, it states that self reported health status was used as an outcome, but previously it was stated that self report studies were excluded. 5. Please provide a rationale for pooling studies with such disparate starting ages, and follow up times. How could this affect results? Can healthy aging be defined consistently across such a broad range? Please show some results stratified by starting age, and follow up time. Or perhaps a meta regression against age or follow up time would be informative. 6. Please also present a rationale for pooling studies with such a broad range of outcomes. The table describing the outcomes should be included in the main document, perhaps as a figure. Other concerns: 7. How was the definition of a longitudinal study operationalized? 8. Increasingly the Paule Mandel estimate of interstudy variance is preferred to that of Der Simonian and Laird, please assess robustness of results to this. 9. The flow diagram says that 5 studies were excluded due to not reporting smoking, alcohol or physical activity. Given that physical activity is not part of the current objectives, how is this reasonable? 10. Please provide the reasons that the 5719 papers were excluded after title/abstract review. 11. For alcohol where light drinking seems to offer a beneficial effect for healthy ageing – please discuss the possibility that this is due to sick people not drinking, or discuss whether this may be a true biologic effect
--	---

VERSION 1 – AUTHOR RESPONSE

Reviewer: 1

Reviewer Name: Sara Hägg

Institution and Country: Karolinska Institutet, Sweden Please state any competing interests or state ‘None declared’: None declared.

Please leave your comments for the authors below

RE: Associations of Smoking and Alcohol Consumption with Healthy Aging – Daskalopoulou et al.

The authors have made a thorough search for publications relating smoking and alcohol consumption to healthy aging. They provide all information needed in order to follow their search strategy and also make all necessary additional statistical tests where applicable. They conclude that non-smokers and also former smokers have a healthier aging trajectory compared to smokers, and that non-smokers are healthier than former smokers. For alcohol consumption, the association is not as easy to explain. The trend though is that drinkers do have a better health profile than non-drinkers.

Some comments:

- Healthy aging is not defined properly. What is meant by healthy aging? Is it survival, measures related to physical function or biomarkers of aging? This has to be clearly stated and should be listed for all studies. If different measures are combined, how is this done?

Thank you very much for your comment.

Our systematic review indicated that healthy ageing, or any other term used as a synonym, was defined in the initial individual studies with considerable heterogeneity. However, all studies used areas that encompassed the concept of being old and maintain a satisfactory functioning. Many of the identified studies (23 out of 28) provided definitions that included physical performance measurements. This finding comes in accordance with another systematic review focusing only on the definition of healthy/successful ageing by Depp and Jeste, 2006. In the supplementary Table A1 we provided the areas of information included in the definition of healthy ageing per study and in addition we now provide a figure (Figure 2) in which the different domains (i.e. physical performance, diseases, mental health, survival, subjective measurements, health status and others -life satisfaction, happiness and pain-) and the number of times which they appear in the 28 eligible studies are illustrated.

- The association between smoking status and healthy aging follows the expected dose-response direction of effect with increased smoking giving a worse health outcome. This conclusion may be supported by many studies and also in terms of causality in Mendelian Randomization studies. Perhaps worth mentioning.
 - The association between alcohol consumption and healthy aging is however more difficult to interpret and does not follow the expected direction of effect. Why would drinkers in all categories be healthier than non-drinkers? This does not make any sense. There are plenty of studies showing this observational association but it is also likely influenced by many confounding factors and reverse causation. In light of this it is extremely important to try to disentangle the relationship better by perhaps adjusting for confounding factors and also look into causal studies using the Mendelian Randomization approach. There is an excellent paper now in the bioRxiv explaining all possible confounding factors and doing a Mendelian Randomization on alcohol consumption and Alzheimer's Disease which is indeed showing a null result; that no such association exist (doi: <https://emea01.safelinks.protection.outlook.com/?url=https%3A%2F%2Fdoi.org%2F10.1101%2F190165&data=01%7C01%7Cchristina.daskalopoulou%40kcl.ac.uk%7C122e8955f4b14ba0a87f08d53d67e789%7C8370cf1416f34c16b83c724071654356%7C0&sdata=rJ%2BymEfQ62CzTgR3DIWOBIVk9OLcpw2iF58468QZeWw%3D&reserved=0>). The current study would benefit from presenting adjusted models and also explain the result better by providing information on possible confounding factors driving these results because they are not likely to be true.
- Thank you very much for both of your comments regarding causality effect and the advantages provided by Mendelian Randomization studies in the avoidance of confounding. Even though, including genotype information would have provided a more comprehensive examination of determinants of healthy ageing, in our systematic review protocol, we only allowed for observational cohort studies examining the associations of smoking and alcohol drinking. As a consequence, studies using genome-wide data were not considered.
- In our meta-analyses, fully-adjusted models were employed and we also provided the confounders that were used for these adjustments (Table 2). Studies were adjusted for a variety of different confounders including: sociodemographic, economic, health behaviour, diseases and physical measurements. However, in our discussion, among other limitations and considerations we also highlighted the fact that our results (i.e. for the association of alcohol drinking and healthy ageing) could be biased by confounders that have not been taken into account, such as health status or former and occasional drinkers to be counted as non-drinkers. For instance, we underlined that in the sub-group of drinkers the positive effect comes mainly from the study of LaCroix et al. (2016); however we have to take into account that in that study former drinkers were also assumed as non-drinkers. We also suggested that more research is required in order to establish or reject the beneficial effects of limited alcohol consumption as in an extended systematic review and meta-analysis of alcohol impact on mortality risk, the benefits of low-volume drinkers largely disappeared once studies were controlled for several design and methodological issues. Finally, our discussion has now been updated with comments about Mendelian Randomization studies (page 16).

Reviewer: 2

Reviewer Name: Julie Byles

Institution and Country: University of Newcastle, Australia Please state any competing interests or state 'None declared': None

Please leave your comments for the authors below An interesting topic of importance - to assess the association between common and significant health risks (smoking and alcohol consumption) and healthy ageing.

Followed appropriate protocols for systematic reviews including PRSIMA, MOOSE, and processes for data extraction and quality assessment with independent raters.

The review was registered in PROSPERO.

73 papers were identified and 28 were included in the review. Most studies were undertaken in USA.

The definition of the risk factors was complicated by:

1) A categorical definition of smoking status, without regard to pack years or time since quitting – this should be discussed.

Thank you very much for your comment; we have now addressed it in page 16.

2) Different measures of alcohol consumption across studies – this has been discussed.

3) Inability to consider ex-drinkers (sick abstainers) – this has been discussed.

The definition of healthy ageing was heterogenous and included:

1) Survival, 12/28

2) Health status 3/28

3) Physical performance 23/28 studies

4) Diseases 18/28

5) Mental health/ Cognition 16/28

6) Life satisfaction 5/28

It is not clear how these very different outcomes were operationalised as “healthy ageing” in the review. It would also be useful to see if effects are different using different operational definitions.

Thank you very much for your comment. In our review, our objective was to identify studies that examined the impact of smoking and alcohol consumption on healthy ageing, no matter how the latter had been defined. As reported, most of the studies, operationalised healthy ageing by measuring difficulties in physical performance, presence of diseases and/or mental health problems. However, there were other studies which defined healthy/successful ageing quite differently. We found that there were some areas not so often included (i.e. life satisfaction) and others that even though operationally different, they were conceptually the same. For instance, in the study of Ford et al., 2009 successful ageing is defined as the ‘sustained independence during the period of observation’. As the authors argue even though this operationalisation seems very different, it encompasses the area of ‘survival’ as well as the areas of disability and cognition since it implies no disability and good cognitive function.

This systematic review confirmed the heterogeneity both in the definition and the measurement of healthy ageing as other systematic reviews have indicated (Peel et al., 2005; Depp and Jeste, 2006; Cosco et al., 2014). Findings from the current study also revealed the urgent need for a unanimous healthy ageing metric that would enable us to make valid comparisons among sub-populations and between different times. Given the heterogeneity in the measurement of smoking and alcohol consumption, it seems quite challenging to assess via a meta-analysis the effects of smoking and alcohol according to the different operational definitions of healthy ageing. Nevertheless, pooled estimates suggested that regardless of the healthy ageing definition, smoking is negatively associated with it and light alcohol consumption exhibits some limited beneficial associations.

The review was also limited by high/moderate bias in the primary studies, particularly due to attrition and confounding. This has been discussed.

27 studies assessed effects of smoking and 18 were included in a meta-analysis. Studies were excluded due to differences in outcome type, or exposure classification.

The results of the meta-analysis show a 2.36 increase odds of healthy ageing (variously defined) for non-smokers compared to current smokers.

21 studies assessed effects of alcohol and 9 were included in a meta-analysis.

The results show drinkers had higher odds of healthy ageing compared to non-drinkers, although the association for moderate drinkers was not statistically significant.

The analyses were not disaggregated to show the separate effects of smoking or alcohol on men and women.

□ Thank you for your comment. In our systematic review, majority of the included studies reported results for a mixed population. More specifically, from the 28 studies identified, only five studies examined men population only and two studies examined women population only. In our meta-analyses, we included two studies that reported results for men only and two that reported results for women only. As a consequence, our data did not allow the examination of the effects of smoking or alcohol consumption separately for men and women. We have commented on this limitation in page 17.

These results are consistent with most of the previous literature and discourse on the effects of smoking and alcohol and the findings from previous studies.

No assessment of population attributable risk for these factors and (un) healthy ageing has been provided (considering prevalence, and changing prevalence, of exposure).

□ Thank you for your insightful comment. To calculate the population attributable risk the estimated prevalence of exposure in the population and the risk ratio for each particular risk factor are needed. However, the information regarding prevalence and changing of prevalence for the exposure variables (i.e. smoking and alcohol consumption) and the outcome variable (i.e. healthy ageing) was challenging to be found and extracted from all the initial studies and for the various categories for which we performed a meta-analysis (smoking: never versus current smokers, never versus former smokers, past/never versus current smokers, never versus past/current smokers; alcohol consumption: drinkers versus non-drinkers, light versus non-drinkers, moderate versus non-drinkers, high versus non-drinkers). We certainly agree that the population attributable risk of these factors and of healthy ageing are important issues to be assessed and as such we feel that this is a limitation of our study.

The potential value of this paper is in summarizing these risks and discussing the interpretation. It is helpful to contrast the summary findings with outliers, as for example the one study finding the negative association between alcohol and healthy ageing. This sort of examination could be drawn out more with implications for both methods, and for understanding differential impacts (on different outcomes) or the effects of dose.

□ Thank you for your comment. In our study we tried to provide aggregated results for the impact of smoking and alcohol consumption on healthy ageing. It is worth-mentioning that even studies that did not participate in our meta-analysis, for instance Tampubolon, 2016, and which gave a healthy ageing definition quite different from the majority of the studies (this is the only study providing a healthy ageing definition based on the measurement of eight biomarkers only) are also in accordance with the aggregated results of our meta-analyses. More specifically, in that study (Tampubolon, 2016) non-smokers compared to current smokers and daily drinkers compared to non-daily drinkers exhibited a better healthy ageing phenotype. The same conclusions were reported to other studies as well that did not participate in our meta-analyses. For instance, Reed et al., 1998, who provided a continuous variable for smoking consumption (Cigarette pack-years = (usual number of cigarettes/day) * (number of years)), reported that the odds for a healthy ageing are significantly decreased as the number of cigarette pack-years increases (manuscript-Table 3). In addition, the positive associations that we found from our meta-analyses of limited alcohol consumption with healthy ageing are in accordance with the results of some initial individual studies. For instance, a beneficial association of limited alcohol consumption is also provided in the study of Hodge, O'Dea et al., 2014 and in the study of Kaplan et al., 2008. The only study which reports a negative association is that of Willcox et al., 2006. However, as already discussed in our manuscript, people who drank more than 3 drinks per day were

compared to non-drinkers only. Hence, a comparison of light or moderate drinkers with non-drinking is not available.

The fact that most studies were conducted in USA and higher income countries should also be discussed. Smoking and alcohol consumption are major issues for countries with emerging economies which are also rapidly ageing.

Thank you for your comment. Our systematic review also indicated the limited research on ageing that has been done in low-and-middle income countries. We found that the majority of the studies has been implemented in high-income countries (24 out of 28 studies). Our findings are in accordance with a previous study indicated the limited research on ageing in low and middle income countries (LMIC), even though by 2050 80% of the people aged 60 years and over will live there. In our discussion and in our conclusion sections, we emphasised the need of future research in this part of the world as estimations show that by 2030 four out of five smoking-related deaths will occur in a LMIC. Finally, taking into account that nowadays 80% of the more than one billion smokers live in LMIC we acknowledged that the economic burden of smoking in those countries is the heaviest.

The conclusions are not strong, and do not reflect the value of the paper. They mainly focus on the lack of consensus in the measures rather than the significance of the findings and how these might be used to promote healthy ageing.

Thank you for your comment. Our conclusions focused on the need of the implementation of a common metric of healthy ageing and of consistent measurement assessments of smoking and alcohol consumption to enable valid comparisons among different groups. However, we also underlined that from our findings smoking abstinence and smoking cessation are positively associated with healthy ageing and should be promoted. As a consequence, we suggested that: "As it takes more than 20 years for most smoking-related diseases to develop, the best practice would be to enforce smoking prevention policies, such as marketing bans and high taxation, and reduce smoking uptake among younger cohorts." Finally, we updated our discussion by suggesting that future research, including also Mendelian Randomization studies and studies allowing for full-confounding, should be prioritised to confirm or not the beneficial association of light alcohol consumption with healthy ageing (page 17-18).

Reviewer: 3

Reviewer Name: Katy Tobin

Institution and Country: Trinity College Dublin, Ireland Please state any competing interests or state 'None declared': None declared

Please leave your comments for the authors below This work aims to combine the results published in longitudinal studies regarding alcohol use and smoking and their effects on healthy ageing. The paper is well written and you have clearly highlighted the difficulties of gathering data on alcohol use and categorising these data, as well as highlighting the need for similar research in low and middle income countries.

I have a few suggestions for minor amendments:

Introduction, line 12: Consider changing "elderly" to "older".

Comment agreed and amended.

Introduction, line 32: Sensitivity to the effects of alcohol, rather than "sensitivity in alcohol" Comment agreed and amended.

Introduction, line 36: Reference 8 is for a book. Please add the specific page number that the reference applies to.

Thank you very much for your comment. This reference is an online book and page numbers are not provided; however, our reference has now been updated with specific chapter number and URL.

Do you mean that the consumption of some amount of alcohol has a different effect depending on age, or that consuming some amount at a young age can create a problem in later years? I expect you mean the former, but the language used is unclear.

Comment agreed and amended.

Introduction, line 39: "older people with drinking problems often do not present to services due to stigma associated with the condition". Consider changing to "older people who drink harmfully often do not present to services due to stigma associated with harmful drinking". Comment agreed and amended.

In my own writing I try to be careful of defining people by an illness that they suffer from, or a behaviour that they demonstrate. Therefore, rather than label a group as "drinkers" or "smokers", I would use the terms "people who drink" and "people who smoke". You might consider this in your work.

Conclusion, line 48: "A positive relationship between limited consumptions of drinking" should be "A positive relationship between limited consumption of alcohol and healthy ageing"

Comment agreed and amended.

Reviewer: 4

Reviewer Name: Abdelmonem A. Affi

Institution and Country: Professor of Biostatistics, Fielding School of Public Health, UCLA, Los Angeles, CA, USA Please state any competing interests or state 'None declared': None

Please leave your comments for the authors below
General comment
The purpose of this study is to conduct a systematic review and meta-analysis of longitudinal studies to synthesize the associations of smoking and alcohol consumption with healthy ageing (HA). To this end, the authors searched major data bases and identified 28 studies for inclusion in their analysis. The statistical analysis produced a pooled effect estimate of various categories of these two risk factors. The authors performed a random effects meta-analysis, and checked heterogeneity and publication bias. They also adjusted for potential publication bias. The authors seem to have used the methods and interpreted the results correctly.

Specific comment

The authors use the term "late survival" without defining it. Please define.

Thank you for your comment. We erroneously used the term "late survival" in the text interchangeably with the term healthy ageing or age in a healthy way. In the revised manuscript this term is not used.

Reviewer: 5

Reviewer Name: Andrea Benedetti

Institution and Country: McGill University, Canada Please state any competing interests or state 'None declared': None declared.

Please leave your comments for the authors below

This is a very interesting paper that tries to meta analyze the associations between smoking and healthy ageing and alcohol and healthy ageing.

I have several concerns, mostly about the heterogeneity across studies.

Confounding related concerns:

1. Studies that were addressing the association between another exposure and healthy ageing but included smoking and alcohol as covariates were included in this meta analysis – were relevant confounders accounted for in these studies?

□ Thank you for your comment. In the meta-analyses of the associations of smoking and alcohol consumption with healthy ageing we used results of the fully adjusted models as these were provided by the initial individual studies. Majority of the studies provided results which were adjusted for sociodemographic and economic confounders (i.e. age, gender, education, income etc.), health behaviour confounders (i.e. physical activity, cigarette smoking, alcohol drinking, hours of sleep etc.), diseases and physical measurements confounders (i.e. heart disease, diabetes, triglyceride, haematocrit, grip strength, etc.) and attitude and social environment confounders (i.e. sense of coherence, financial stress, social participation etc.). The confounders that were used per study are provided in aggregated categories in Table 2.

2. More information on whether it is believed that the studies contributing results for alcohol were adequately adjusted for smoking and other important confounders is necessary. Similarly for those reporting a smoking-outcome association. The list of variables provided makes it difficult to assess if the right variables were adjusted for. I would like to know the proportion believed to be adequately adjusted. Perhaps stratifying if there are many that are not adequately adjusted would be appropriate.

□ Thank you very much for your insightful comment. We used the QUIPS tool (Quality assessment of the studies) to assess the quality of our studies. We quote from the manuscript: "During the application of the QUIPS tool smoking and/or alcohol consumption were considered as the only prognostic factors and all other variables, used as explanatory variables of the model, were considered as confounders." Consequently, for each study, we assessed if important potential confounders were appropriately accounted for, limiting potential bias with respect to the relationship between the prognostic factors (smoking or alcohol consumption) and the outcome (healthy ageing). To conclude about the quality of the studies regarding confounding, the two independent reviewers performing the quality assessment checked the following:

1. Are all important confounders measured?
2. Are clear definitions of the important confounders measured?
3. Is the measurement of all important confounders adequately valid and reliable?
4. Are the method and setting of confounding measurement the same for all study participants?
5. Have appropriate methods been used if imputation was used for missing confounder data?
6. Are important potential confounders accounted for in the study design?
7. Are important potential confounders accounted for in the analysis?

After replying to these seven questions reviewers provided their assessment regarding confounding bias. In the supplementary Table A2, we provided quality assessment results per study. We concluded that from the 28 studies in the "confounding assessment domain", 19 had moderate risk of bias, 9 had low risk of bias and no study was characterised as having high risk of bias. "Regarding the quality assessment of the studies, limited disagreement among the six different domains per study was reached between the two reviewers, who independently assessed them, concluding that the quality assessment tool was reliable and did not allow great amount of misjudgement". As a consequence, we feel confident that our results were adequately adjusted. However, as most of the studies exhibited moderate bias in this domain we recommended that future research should not neglect this issue and consider design of studies including important confounders during data collection and fitting of models.

Furthermore, we also examined if important confounders had been taken into account to the adjusted results participated in the meta-analyses of smoking and alcohol consumption. On average, 75% of important confounders were included in the models used for a meta-analysis of smoking associations with healthy ageing and 81% of important confounders were included in the models used for the meta-analysis of alcohol consumption (please see tables below).

Confounders included in studies participated in meta-analyses (smoking)

Study	Age	gender	education	alcohol consumption			physical activity physical illness/morbidity Total		
Bell et al., 2014	x		17%						
Britton et al., 2008			x	x	33%				
Ford et al., 2000			x	x	x	x	x	100%	
Gu et al., 2009	x	x	x	x	x	x	100%		
Guralnik & Kaplan, 1989			x	x	x	x	67%		
Gureje et al., 2014	x	x	x	x	x	x	100%		
Hamer et al., 2013	x	x	x	x	67%				
Hodge, English et al., 2013			x	x	x	x	x	x	100%
Hodge, O'Dea et al., 2014			x	x	x	x	x	x	100%
Kaplan et al., 2008	x	x	x	x	x	x	100%		
LaCroix et al., 2016	x	x	x	x	67%				
Newson et al., 2010	x	x	x	x	x	83%			
Pruchno & Wilson-Genderson, 2015			x	x	x	x	x	x	100%
Sabia et al., 2012	x	x	x	x	x	83%			
Sarnak et al., 2008	x	x	x	50%					
Shields & Martel, 2006	x	x	x	x	x	x	100%		
Terry et al., 2005	x	x	x	x	67%				
Willcox et al., 2006	x		17%						

Confounders included in studies participated in meta-analyses (alcohol)

Study	age	gender	education	smoking	physical activity physical illness/morbidity Total			
Britton et al., 2008			x	x	33%			
Ford et al., 2000			x	x	x	x	100%	
Gu et al., 2009	x	x	x	x	x	x	100%	
Guralnik & Kaplan, 1989			x	x	x	x	67%	
Gureje et al., 2014	x	x	x	x	x	x	100%	
Hodge, English et al., 2013			x	x	x	x	x	100%
Hodge, O'Dea et al., 2014			x	x	x	x	x	100%
LaCroix et al., 2016	x	x	x	x	67%			
Sun et al., 2011	x	x	x	x	67%			

Outcome related concerns:

3. The outcome definition is difficult for me. What is "healthy ageing"? You exclude studies where this was self report – how was it assessed in the other studies? Similarly around the idea that if this was reported as multiple outcomes. More detail is needed here.

4. In the results, it states that self reported health status was used as an outcome, but previously it was stated that self report studies were excluded.

□ Thank you for both of your comments regarding the healthy ageing outcome. We excluded studies reporting healthy ageing as multiple-outcome. For instance, the study of Zaslavsky et al., 2014 entitled: "Trajectories of positive aging: observations from the women's health initiative study" was a potential eligible study. However, during the full-text screening process we concluded that this study was not eventually eligible to be included in our systematic review as it examined two different dimensions/outcomes of healthy ageing called: physical-social functioning and emotional functioning. We also excluded studies that used a definition of healthy ageing which was solely self-reported (for instance studies in which healthy ageing was defined by asking participants about their health status). This does not mean that we excluded studies that had also self-reported areas of information in the definition of healthy ageing.

5. Please provide a rationale for pooling studies with such disparate starting ages, and follow up times. How could this affect results? Can healthy aging be defined consistently across such a broad range? Please show some results stratified by starting age, and follow up time. Or perhaps a meta regression against age or follow up time would be informative.

Thank you very much for your comments. As we expected heterogeneity we used random-effects meta-analysis to produce a pooled effect estimate, which is also advocated when pooling observational data according to the MOOSE guidelines. To estimate the between-study variability we used the DerSimonian-Laird (DL) estimator and we also performed sensitivity analysis by using the less biased estimator Paule-Mandel (PM). The highest heterogeneity was 87.2% from the DL estimator and 81.3% from the PM estimator (please see comment 8). As a consequence, even though initial study designs were different, heterogeneity remained within acceptable limits to allow us to perform a meta-analysis and provide aggregated results. In addition, we tried to have as much as possible homogeneous groups with regards of the exposure variables; this is the reason for which we performed various sub-group meta-analyses. Nevertheless, as discussed in our manuscript, the number of studies included in each of our meta-analysis was limited not allowing us to perform meta-regression and assess the impact of different baseline age and follow-up. As our aim was a quantification of the relationship of smoking and healthy ageing and of alcohol consumption and healthy ageing, and as heterogeneity remained within acceptable limits, we pooled our initial studies together.

6. Please also present a rationale for pooling studies with such a broad range of outcomes. The table describing the outcomes should be included in the main document, perhaps as a figure.

Thank you very much for your comment. In our studies there was only one outcome: healthy ageing. This has been defined in the initial studies by including various areas of information. We have now included a figure (figure 2-page 10) which depicts the domains identified in the healthy ageing definition and the number of times that these domains were identified in the 28 eligible studies. Nevertheless, the lack of a common definition of healthy ageing has been acknowledged as a limitation of our study. However, we feel that the inclusion of an aggregated effect estimate contributes to a quantitative interpretation of our findings and hence provides more information compared to a narrative report.

Figure 2: Areas of information included in the definition of healthy ageing

Other concerns:

7. How was the definition of a longitudinal study operationalized?

Thank you for your comment. We included observational studies/cohort studies which assessed exposures in baseline assessment and assessed healthy ageing in a subsequent wave.

8. Increasingly the Paule Mandel estimate of interstudy variance is preferred to that of Der Simonian and Laird, please assess robustness of results to this.

Thank you for your comment. By using the 'metafor' package of R statistical software we reproduced pooled results by using the Paule-Mandel estimate. As shown in the table below robustness of our results is confirmed by using these two different estimators of the between-study variance. This table is also provided in the revised manuscript as Supplementary Material: Table A3. docx (Sensitivity analysis of the meta-analytic results) (page 17-18)

Analysis	Number of Studies	ORs and 95%CI	p-value I2	ORs and 95%CI
Never vs Current Smokers 23.6%	7	2.36 (2.03-2.75)	<0.001 43.3%	2.29 (2.03-2.59) <0.001

Never vs Former Smokers 50.0%	5	1.32 (1.23-1.41)	<0.001	32.8%	1.31 (1.21-1.43)	<0.001
Past/Never vs Current	6	1.72 (1.20-2.47)	0.003	87.2%	1.69 (1.25-2.29)	<0.001
Never vs Past/Current	5	1.29 (1.16-1.43)	<0.001	0.0%	1.29 (1.16-1.43)	<.001
0.0%						
Drinkers vs Non-Drinkers 68.2%	5	1.28 (1.08-1.52)	0.004	72.1%	1.29 (1.10-1.50)	0.002
Light vs Non-Drinkers	3	1.12 (1.03-1.22)	0.010	0.0%	1.12 (1.03-1.22)	0.010
Moderate vs Non- Drinkers 77.0%	4	1.35 (0.93-1.97)	0.112	71.4%	1.37 (0.90-2.08)	0.138
High to Non-Drinkers	3	1.25 (1.09-1.44)	0.002	0.0%	1.25 (1.09-1.44)	0.002
0.0%						

9. The flow diagram says that 5 studies were excluded due to not reporting smoking, alcohol or physical activity. Given that physical activity is not part of the current objectives, how is this reasonable?

Thank you for your comment. As part of a larger body of work considering modifiable lifestyle factors and healthy ageing, we originally planned to carry out a systematic review focusing on the following: physical activity, smoking and alcohol consumption. The current systematic review specifically focuses on the findings related to smoking, alcohol consumption and healthy ageing outcomes since a sufficient amount of literature was identified on this topic alone. Our findings regarding healthy ageing and physical activity have been presented elsewhere (Daskalopoulou et al., 2017). As a consequence, our PRISMA flowchart included some information on physical activity also. We have now revised our PRISMA flowchart.

10. Please provide the reasons that the 5719 papers were excluded after title/abstract review.

Thank you very much for your comment. We excluded records after title/abstract review for the following reasons:

- books, conference papers, dissertations
- cross-sectional studies, animal studies
- hospitalised participants or participants belonging to a specific clinical population

Our exclusion criteria were pre-specified in the PROSPERO protocol.

11. For alcohol where light drinking seems to offer a beneficial effect for healthy ageing – please discuss the possibility that this is due to sick people not drinking, or discuss whether this may be a true biologic effect

Thank you for your comment. In our discussion we commented that relating moderate alcohol consumption with health benefits should not come without question since there are many other issues to consider. Our results could be biased by confounders that have not been taken into account, such as health status or former and occasional drinkers to be counted as non-drinkers. In addition, we have now updated our discussion by including comments regarding the true biologic effect (page 16).

Additional References

- Daskalopoulou C, Stubbs B, Kralj C, et al. Physical activity and healthy ageing: A systematic review and meta-analysis of longitudinal cohort studies. *Ageing Res Rev* 2017;38:6-17.
- Depp CA, Jeste DV. Definitions and predictors of successful aging: a comprehensive review of larger quantitative studies. *Am J Geriatr Psychiatry* 2006;14(1):6-20.
- Cosco TD, Prina AM, Perales J, et al. Operational definitions of successful aging: a systematic review. *International psychogeriatrics* 2014;26(3):373-81.
- Peel NM, McClure RJ, Bartlett HP. Behavioral determinants of healthy aging. *American journal of preventive medicine* 2005;28(3):298-304.

Zaslavsky O, Cochrane BB, Woods NF, et al. Trajectories of positive aging: observations from the women's health initiative study. *International psychogeriatrics* 2014;26(8):1351-62. doi: 10.1017/s1041610214000593

VERSION 2 – REVIEW

REVIEWER	Andrea Benedetti McGill University Canada
REVIEW RETURNED	04-Jan-2018

GENERAL COMMENTS	Most comments were satisfactorily addressed. However, in my initial review I asked the following: Please provide a rationale for pooling studies with such disparate starting ages, and follow up times. How could this affect results? Can healthy aging be defined consistently across such a broad range? Please show some results stratified by starting age, and follow up time. Or perhaps a meta regression against age or follow up time would be informative. Your response, highlighting the estimates of I-squared obtained, while interesting, does not address my comment.
---

REVIEWER	Sara Hägg Karolinska Institutet, Sweden
REVIEW RETURNED	08-Jan-2018

GENERAL COMMENTS	No further comments.
----------------------

REVIEWER	Julie Byles University of Newcastle, Australia
REVIEW RETURNED	22-Jan-2018

GENERAL COMMENTS	The authors have addressed the many comments made by the reviewers. Many of the criticisms cannot be “corrected” as they relate to limitations in the source data. However these are discussed. A major limitation remains the heterogeneity of the outcome “healthy ageing”. Figure 2 lists the outcomes, but probably doesn’t help understand the multi-dimensional operational definitions that were used in different studies. For example studies would not have used “survival” as the only criterion. And surely all studies included survival (ie. Being alive). Excluded studies that reported on other valid outcomes such as healthy years (for example) are considered in the discussion. Authors have included a reference to the more recent studies using Mendelian Randomisation. I was not able to find the discussion concerning the inability to
--

	undertake comparison of effects by gender (said to be on page 17). The importance of the LMIC isn't so much that the results aren't generalizable, but rather that the effects may be also of increasing importance in these countries as the prevalence of smoking and alcohol use increases. This point is made in the discussion, but lost in the dot points. Item 6 on definition of outcomes relates to lack of consistent definition of healthy ageing, which limits the use of a homogenous outcome for this review. This has been discussed by authors and reviewers. OTHER COMMENTS ON REVISED MANUSCRIPT: The authors have addressed the many comments made by the reviewers. Many of the criticisms cannot be "corrected" as they relate to limitations in the source data. However these are discussed. A major limitation remains the heterogeneity of the outcome "healthy ageing". Figure 2 lists the outcomes, but probably doesn't help understand the multi-dimensional operational definitions that were used in different studies. For example studies would not have used "survival" as the only criterion. And surely all studies included survival (ie. Being alive). Excluded studies that reported on other valid outcomes such as healthy years (for example) are considered in the discussion. Authors have included a reference to the more recent studies using Mendelian Randomisation. I was not able to find the discussion concerning the inability to undertake comparison of effects by gender (said to be on page 17). The importance of the LMIC isn't so much that the results aren't generalizable, but rather that the effects may be also of increasing importance in these countries as the prevalence of smoking and alcohol use increases. This point is made in the discussion, but lost in the dot points.
--	---

REVIEWER	Katy Tobin Trinity College Dublin Ireland
REVIEW RETURNED	22-Jan-2018

GENERAL COMMENTS	I am happy that the authors have amended the manuscript based on the comments I submitted previously.
---

VERSION 2 – AUTHOR RESPONSE

Response to Reviewers

Reviewer: 1

Reviewer Name: Sara Hägg

Institution and Country: Karolinska Institutet, Sweden Please state any competing interests or state 'None declared': None declared

Please leave your comments for the authors below No further comments.

Reviewer: 2

Reviewer Name: Julie Byles

Institution and Country

Please state any competing interests or state 'None declared': NO competing interests

Please leave your comments for the authors below Item 6 on definition of outcomes relates to lack of consistent definition of healthy ageing, which limits the use of a homogenous outcome for this review. This has been discussed by authors and reviewers.

OTHER COMMENTS ON REVISED MANUSCRIPT:

The authors have addressed the many comments made by the reviewers. Many of the criticisms cannot be "corrected" as they relate to limitations in the source data. However these are discussed.

A major limitation remains the heterogeneity of the outcome "healthy ageing". Figure 2 lists the outcomes, but probably doesn't help understand the multi-dimensional operational definitions that were used in different studies. For example studies would not have used "survival" as the only criterion. And surely all studies included survival (ie. Being alive).

➤ Thank you for your comment. The different categories that were identified in the definition of healthy ageing are listed in Figure 2. Analytical information showing the multi-dimensional definitions per study is provided in the "Supplemental Table A1". For example "survival", identified in 12 studies, was used together with the category of "physical performance" and/or "diseases" and/or "mental health". Lastly, we have now updated our discussion regarding the fact that survival was assumed in the definition of healthy ageing, even though this was not explicitly stated (page 17).

Excluded studies that reported on other valid outcomes such as healthy years (for example) are considered in the discussion.

Authors have included a reference to the more recent studies using Mendelian Randomisation.

I was not able to find the discussion concerning the inability to undertake comparison of effects by gender (said to be on page 17).

➤ Thank you for your comment. Our comment on page 17 was about the limitation of our study to perform meta-regression analyses for various confounders and assess their impact. We quote: "performing meta-regression analysis and assessing the confounding effect of covariates was not possible either". To highlight this we now mention specific covariates to which we refer (i.e. gender).

The importance of the LMIC isn't so much that the results aren't generalizable, but rather that the effects may be also of increasing importance in these countries as the prevalence of smoking and alcohol use increases. This point is made in the discussion, but lost in the dot points.

➤ Thank you for your comment. In the dot points of our manuscript (i.e. "Strengths and Limitations of this study", page 3) we have not commented in the increased prevalence of smoking and alcohol use in LMICs, as we feel that this is not a strength or a limitation of our study. On the contrary, a limitation of our study is the limited research/data from LMICs besides the fact that by 2050 more than 80% of

people 60 years old will reside there. As a consequence, we commented that our results cannot be generalisable as they are mostly based on research that has been conducted in non LMICs. However, as we also feel that the high prevalence of smoking and the increased use of alcohol are of considerable importance, we updated our discussion (page 18) highlighting even more those problems.

Reviewer: 3

Reviewer Name: Katy Tobin

Institution and Country: Trinity College Dublin, Ireland Please state any competing interests or state 'None declared': None declared

Please leave your comments for the authors below I am happy that the authors have amended the manuscript based on the comments I submitted previously.

Reviewer: 5

Reviewer Name: Andrea Benedetti

Institution and Country: McGill University, Canada Please state any competing interests or state 'None declared': None declared.

Please leave your comments for the authors below

Most comments were satisfactorily addressed. However, in my initial review I asked the following: Please provide a rationale for pooling studies with such disparate starting ages, and follow up times. How could this affect results? Can healthy aging be defined consistently across such a broad range? Please show some results stratified by starting age, and follow up time. Or perhaps a meta regression against age or follow up time would be informative.

Your response, highlighting the estimates of I-squared obtained, while interesting, does not address my comment.

➤ Thank you very much for your comment. In our study we have performed eight different meta-analyses to quantify the associations of smoking with healthy ageing and of alcohol consumption with healthy ageing. More explicitly, for smoking we have performed the following four meta-analyses: never versus current smoking, never versus former smoking, former/never versus current smoking and never versus former/current smoking; whereas for alcohol consumption, we have performed the following four meta-analyses: drinkers versus never drinkers, less than one drink per day versus non-drinkers, one-two drinks per day versus non-drinkers and two-four drinks per day versus non-drinkers. Our data did not allow the examination of stratified (per age and follow-up time) analyses for each of the aforementioned meta-analyses. However, we were able to produce some stratified analyses which are presented below and in the updated manuscript.

➔ Age stratification

Smoking and Healthy Ageing

Studies were stratified according to their baseline mean age; studies with baseline mean age more than 65 years old and studies with baseline mean age less than or equal to 65 years old.

The association of smoking with healthy ageing did not change direction or strength when we stratified our data per baseline mean age (Figure 1). However, we did observe that the beneficial

association of non-smoking was higher in studies with baseline mean age less than 65 years old. This finding is in accordance with other studies examining smoking and adverse health outcomes in old populations (Herman et al, 2008; Chang et al., 2012). When studies incorporate an older sample this is biased in favour of people who smoke and survive compared to smokers who die (survival bias). The only association that differed once we stratified per baseline mean age was that in the analysis of former/never compared to current smokers. Studies with baseline mean age more than 65 years old showed a non-significant association (OR: 1.34, 95%CI: 0.65-2.76) whereas studies with baseline mean age less than 65 years old showed a statistically significant association (OR:1.90, 95%CI:1.51-2.40). The non-significant result in the group of former/never smokers compared to current smokers had also been highlighted and commented in the discussion part (page 14) of our manuscript, as it was revealed when we performed the trim-and-fill algorithm analysis.

Figure 1: Smoking and healthy ageing-age stratified

Alcohol Consumption and Healthy Ageing

We were not able to perform age-stratified analyses for the associations of alcohol consumption (drinkers, light, moderate, high drinkers compared to non-drinkers) with healthy ageing as studies were homogeneous regarding their baseline mean age.

➔ Follow-up stratification

Smoking and Healthy Ageing

Studies were stratified based on their follow-up time; studies with follow-up time more than 10 years and studies with follow-up time less than or equal to 10 years.

We were able to do follow-up stratified analyses for two smoking-related meta-analyses (former/never versus current and never versus former/current). Stratification was not possible to the other two smoking related meta-analyses (never versus current and never versus former) as all studies had follow-up time more than 10 years.

Follow-up stratification also revealed that in the group of former/never smokers compared to current smokers the association was not significant in studies with less than 10 years follow-up (OR: 1.41, 95%CI: 0.99-2.01). We did not observe any considerable differences in the meta-analysis of never smoking compared to former/current smoking with healthy ageing once we stratified per follow-up time.

Figure 2: Smoking and healthy ageing-follow-up time stratified

Alcohol consumption and Healthy Ageing

We were able to assess the impact of follow-up time in one meta-analysis of alcohol consumption and healthy ageing. More specifically, we were able to provide follow-up stratified results in the meta-analysis of drinkers versus non-drinkers. All other meta-analyses (<1 drink per day, 1-2 drinks per day, 2-4 drinks per day compared to non-drinkers) were homogeneous regarding their follow-up time (i.e. more than 10 years).

The association of alcohol consumption (drinkers compared to never-drinkers) with healthy ageing was significantly positive in studies with follow-up time more than 10 years (OR:1.41, 95%CI: 1.36-1.48) whereas studies with follow-up time less than 10 years showed a non-significant beneficial association (OR: 1.09, 95%CI: 0.95-1.24). The pooled result revealed a beneficial association

(OR:1.28, 95%CI:1.36-1.48). Follow-up stratification confirmed that the significantly positive effect in this alcohol-group meta-analysis comes from the study of LaCroix et al. 2016. In our discussion part of the manuscript (page 16), we have underlined that the beneficial association of drinkers versus non-drinkers has to be interpreted with extra caution as it comes mainly from one study. In addition, we have also emphasised the fact that in that study (LaCroix et al., 2016) former drinkers were also assumed as non-drinkers.

Figure 3: Alcohol and healthy ageing-follow-up stratified

Conclusions

The age and follow-up stratified analyses showed that our pooled results were in accordance with the results of the stratified analyses. Even though studies had disparate starting ages and follow-up time, pooled effect estimates were not severely affected in direction or in strength by the former. Meta-analytical results which require extra caution (i.e. former/never smokers compared to current smokers and drinkers compared to non-drinkers) have been underlined and commented in the manuscript. All things considered, and the fact that heterogeneity I^2 remained within acceptable limits, we decided to perform a meta-analysis and provide pooled effect estimates assisting to a quantitative synthesis of our findings.

References

1. Hernan MA, Alonso A, Logroscino G. Cigarette smoking and dementia: potential selection bias in the elderly. *Epidemiology (Cambridge, Mass)* 2008;19(3):448-50. doi: 10.1097/EDE.0b013e31816bbe14
2. Chang C-CH, Zhao Y, Lee C-W, et al. Smoking, death, and Alzheimer's disease: A case of competing risks. *Alzheimer disease and associated disorders* 2012;26(4):300-06. doi: 10.1097/WAD.0b013e3182420b6e